# Suppression without inhibition: how retinal computation contributes to saccadic suppression

Saad Idrees [1,2,10], Matthias-Philipp Baumann[1,3], Maria M. Korympidou[1,2,4], Timm Schubert [1,4], Alexandra Kling[5], Katrin Franke[4,6], Ziad M. Hafed [1,3], Felix Franke [7,8,9✉] & Thomas A. Münch [1,4✉]

Visual perception remains stable across saccadic eye movements, despite the concurrent strongly disruptive visual flow. This stability is partially associated with a reduction in visual sensitivity, known as saccadic suppression, which already starts in the retina with reduced ganglion cell sensitivity. However, the retinal circuit mechanisms giving rise to such suppression remain unknown. Here, we describe these mechanisms using electrophysiology in mouse, pig, and macaque retina, 2-photon calcium imaging, computational modeling, and human psychophysics. We find that sequential stimuli, like those that naturally occur during saccades, trigger three independent suppressive mechanisms in the retina. The main mechanism is triggered by contrast-reversing sequential stimuli and originates within the receptive field center of ganglion cells. It does not involve inhibition or other known suppressive mechanisms like saturation or adaptation. Instead, it relies on temporal filtering of the inherently slow response of cone photoreceptors coupled with downstream non-linearities. Two further mechanisms of suppression are present predominantly in ON ganglion cells and originate in the receptive field surround, highlighting another disparity between ON and OFF ganglion cells. The mechanisms uncovered here likely play a role in shaping the retinal output following eye movements and other natural viewing conditions where sequential stimulation is ubiquitous.

[1] Werner Reichardt Centre for Integrative Neuroscience, University of Tübingen, 72076 Tübingen, Germany. [2] International Max Planck Research School, University of Tübingen, 72074 Tübingen, Germany. [3] Hertie Institute for Clinical Brain Research, University of Tübingen, 72076 Tübingen, Germany. [4] Institute for Ophthalmic Research, University of Tübingen, 72076 Tübingen, Germany. [5] Department of Neurosurgery, Stanford School of Medicine, Stanford, CA 94305, USA. [6] Bernstein Center for Computational Neuroscience, University of Tübingen, 72076 Tübingen, Germany. [7] Bio Engineering Laboratory, ETH Zürich, 4058 Basel, Switzerland. [8] Institute of Molecular and Clinical Ophthalmology Basel, 4031 Basel, Switzerland. [9] Faculty of Science, University of Basel, 4056 Basel, Switzerland. [10] Present address: Center for Vision Research, York University, Toronto, ON M3J 1P3, Canada. ✉email: felfranke@googlemail.com; thomas.muench@gmail.com

Vision appears as a continuous and coherent process. This is a striking achievement of the visual system, considering that the visual flow across the retina is not continuous, but governed by frequent and sudden changes, irregularities, and disruptions. As a consequence of this active vision, or the process of active exploration of the visual environment, the meaningful images falling onto the retina are only brief snapshots of the world, interrupted by blinks and rapid motion. The most prominent cause of such disruptions are eye movements. Saccades, for example, are critical for efficiently sampling the visual world[1–3], which is particularly true for species in which high visual resolution is limited to a small fraction of the overall visual space, such as the foveal region in primates. On the other hand, as a result of saccades, the number of photons falling onto a given area of the retina can change by several orders of magnitude within tens of milliseconds, causing sudden and frequent visual transients of local intensity across the entire retina. From the perspective of the retina, saccades are therefore equivalent to strong visual stimuli, and they are a powerful model for a very profound question of visual neuroscience: how does the visual system extract robust information from the meaningful snapshots of the world, in the face of frequent, strong, and disruptive other input?

Perceptually, saccadic disruptions are minimized by reducing the sensitivity of the visual system to new input around the time of saccades—a phenomenon known as saccadic suppression. While this phenomenon has been extensively characterized over the past few decades[4–11], its underlying mechanisms still remain unclear. Several electrophysiological studies have shown neural correlates of saccadic suppression throughout the visual system, namely a modulation of neural activity and/or sensitivity around the time of saccades[5,10–15]. These observations have often been interpreted to be caused by active suppressive signals originating from (pre-) motor areas, such as corollary discharge signals related to the saccadic eye-movement command[6,10,16–18]. Most studies investigating the mechanisms of saccadic suppression have therefore focused on cortical or subcortical neuronal recordings and/or on behavioral measures of perceptual state, largely neglecting the consequence of visual processing in early visual pathways, for example in the retina.

The retina is an independent signal processing front end in the visual system, before visual information is sent along the optic nerve to higher brain areas. Consequently, image processing triggered by visual transients, such as those that naturally occur during active vision, including saccades, could potentially lead to altered retinal output. Retinal signal processing could therefore contribute to perceptual saccadic suppression. Some studies have investigated how the retina processes information in the context of spatiotemporal dynamics that occur during natural visual behavior[19–31], including saccades[32–38]. A retinal neural correlate of perceptual saccadic suppression has recently been shown by a previous study from our labs[4]. There, we showed that the retinal output is indeed altered by saccade-like image shifts. In most mouse and pig retinal ganglion cells (RGCs) that we recorded from, responses to brief probe flashes were suppressed when preceded by saccade-like image displacements across the retina. This retinal saccadic suppression had properties consistent with the perceptual suppression of probe flashes reported by human subjects using similar images, and following either real or simulated saccades. In fact, we observed elementary properties of perceptual saccadic suppression, such as its dependency on background scene statistics, already at the level of the retinal output, providing strong evidence of a retinal mechanism directly contributing to perceptual saccadic suppression.

In this study, we describe such a mechanism. We experimentally mimicked the visual flow resulting from saccades and recorded the neural activity of the output neurons of the retina (RGCs) from ex vivo retinae of mice, pigs, and macaque monkeys. We found that retinal saccadic suppression was the result of multiple mechanisms, the most significant of which was a specific visual processing motif within an RGC's receptive field center. This motif, which we call dynamic reversal suppression, did not depend on any inhibitory signals; it resulted from temporal filtering of inherently slow cone photoreceptor responses coupled with nonlinearities in the downstream retina pathways. Two further components of suppression originated from beyond the RGC's receptive field center, only one of them driven by GABAergic inhibition. Interestingly, these two additional components were observed primarily in ON RGCs, highlighting yet another disparity between ON and OFF type RGCs. Perhaps one of the most intriguing outcomes of this study, also consistent with observations of perception[4], is that the suppressive effects observed in RGCs were not exclusively triggered by saccades, but occurred for many scenarios involving sequential visual stimulation, which are ever-present during natural vision. Therefore, while the results described here are crucial for understanding the mechanisms of saccadic suppression, they also elucidate more general mechanisms of retinal signal processing across any time-varying visual input over short time scales (10–1000 ms).

## Results

**Experimental approach**. We measured the modulation of retinal ganglion cell (RGC) output following saccade-like changes of the visual input with a variety of different light stimulation strategies (Fig. 1a and Supplementary Fig. 1). The basic experimental paradigm was similar to that described in[4]. Briefly, we recorded spiking activity of RGCs in isolated ex vivo mouse retinae using both high-density and low-density multielectrode arrays (MEAs). Each retina was exposed to a background texture having one of several possible spatial scales that defined its spatial spectrum (fine to coarse, "Methods", Supplementary Fig. 2). We simulated saccade-like image displacements by rapidly translating the texture globally across the retina ("Methods"; Fig. 1a). Most RGCs responded robustly to such saccade-like texture displacements (see Fig. 1b for responses of example ON and OFF RGCs). At different times relative to the saccade-like texture displacements (saccades from now on), we presented a brief probe flash (Fig. 1c). We then analyzed how the response (spike rate of the RGC) to this probe flash was influenced by the preceding saccade, by comparing it to the response to the flash presented in isolation (baseline). To quantify RGC response modulation, we calculated a modulation index ("Methods") which quantified how much a cell's flash response was modulated by a temporally close saccade. We first isolated the flash-induced response component by subtracting the saccade-only response (e.g., Fig. 1b) from the response to the composite saccade-flash stimulus (e.g., Fig. 1c). Based on this flash-induced response component (Fig. 1d), we calculated the modulation index as $(r_d - r_b)/(r_d + r_b)$. Here, $r_d$ is the peak response to the probe flash presented with a delay $d$ relative to saccade onset, and $r_b$ is the baseline (peak response to the flash presented ~2 s after the saccade). This modulation index is negative when flash-induced responses are suppressed (Fig. 1d shows, on the horizontal dashed line, the example cells' modulation indices for the responses at each flash time). In yet further recordings, we applied various manipulations to this base paradigm to probe for the mechanisms underlying modulation of RGC responses following saccades. To generalize our findings across other species, we also performed similar analyses of pig and macaque RGC data.

## Similarities and differences in retinal saccadic suppression across ON and OFF type RGCs. Suppression was robust across

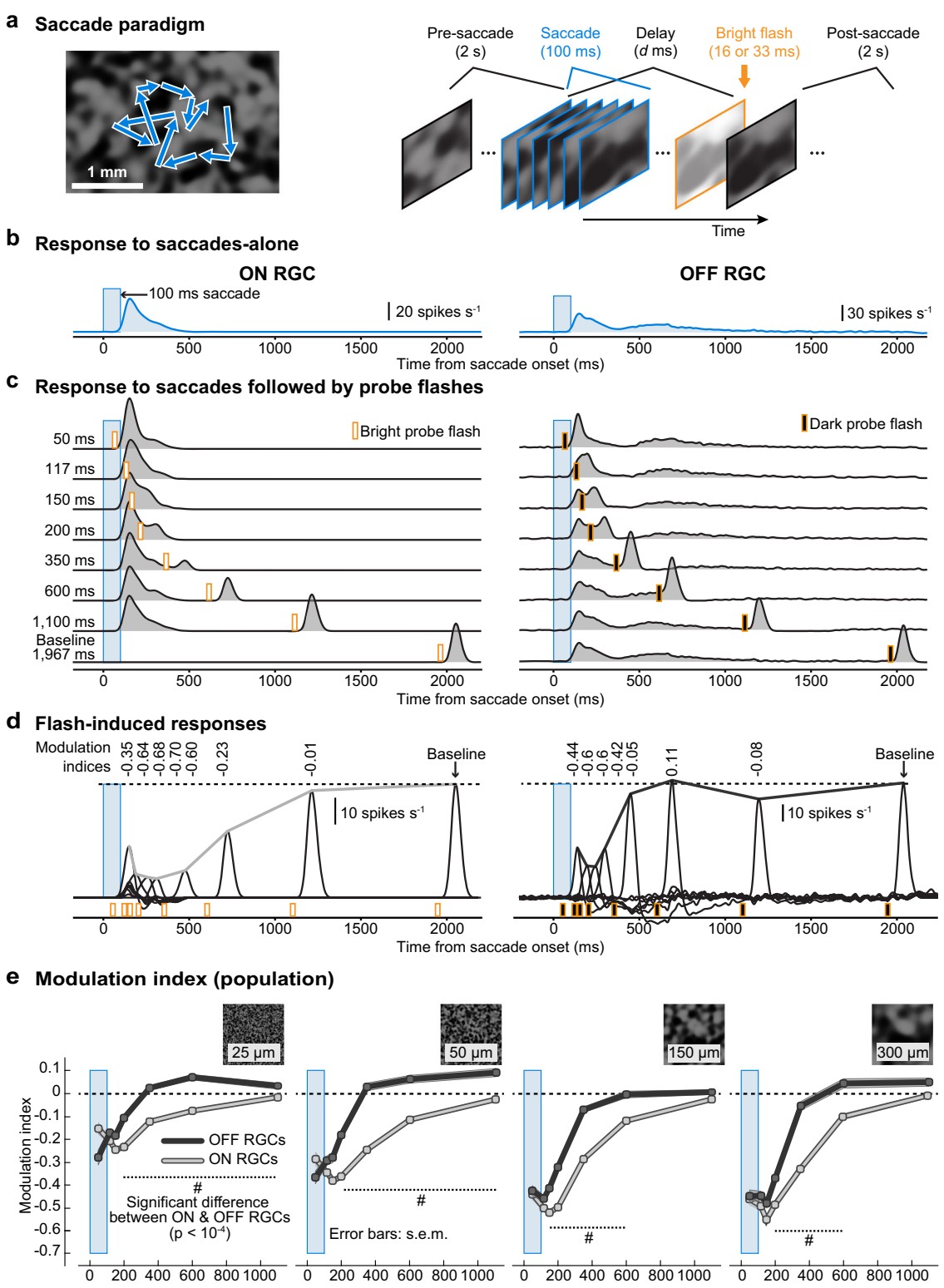

**a  Saccade paradigm**

**b  Response to saccades-alone**

ON RGC                    OFF RGC

**c  Response to saccades followed by probe flashes**

**d  Flash-induced responses**

**e  Modulation index (population)**

most RGCs that we recorded from, consistent with what we reported previously[4]. Here, we more closely inspected functionally different RGCs. Specifically, throughout this study, we divided RGCs into ON and OFF types (i.e., into RGCs responding best to light increments or decrements, respectively; Methods). Unless otherwise noted, we always quantified the modulation index defined above for ON RGCs based on their responses to bright probe flashes and for OFF RGCs based on their responses

to dark probe flashes (Fig. 1c, d). Flash responses following a saccade were suppressed in both ON and OFF RGCs, as seen in Fig. 1d for two example cells. Figure 1e shows the temporal profile of the mean population modulation index for ON and OFF cells, and Supplementary Fig. 3 the underlying population data. Suppression was consistently stronger for coarser background textures (Fig. 1e and Supplementary Fig. 3), for both ON and OFF RGCs. This is consistent with[4], where we showed that this

**Fig. 1 Similarities and differences in retinal saccadic suppression across ON and OFF RGCs. a** RGC action potentials were recorded from ex vivo retinae placed on multielectrode arrays. Saccades were mimicked by displacing a texture projected onto the retina (blue arrows in the left panel indicate texture-displacement paths). The texture remained static for 2 s and was then displaced over 100 ms (blue outlines) followed by a brief probe flash (here, a bright probe flash is depicted, orange outline). Each trial consisted of 39 such successive saccade-flash sequences (Supplementary Fig. 1a). **b, c** Average activity (firing rate) of an example ON RGC (left column) and OFF RGC (right column) to 39 saccade sequences not followed by a probe flash (**b**), and to 39 saccade sequences followed by probe flashes at different delays after saccade onset (**c**). Blue window: timing of saccades; orange markers: timing of probe flashes. **d** Isolated flash-induced responses (firing rate) of the same RGCs obtained by subtracting responses to saccades-alone (**b**) from responses to saccades followed by probe flashes (**c**). Lines connecting the response peaks highlight the time courses of retinal saccadic suppression relative to baseline flash-induced responses. Numbers above each response peak represent the modulation index which quantifies how much the probe flash response is modulated by the preceding saccade ("Methods", negative modulation indices correspond to suppressed flash-induced responses). **e** Population modulation index (mean ± s.e.m.) of ON (light gray) and OFF (dark gray) RGCs, for different background textures with different spatial scales (left to right: fine to coarse). The number of ON and OFF RGCs in the population varied between 68 and 574 for different flash times and textures (see Supplementary Fig. 3 for exact numbers and relevant statistics). Hash symbols: significant modulation difference between ON and OFF RGCs ($P < 10^{-4}$, two-tailed Wilcoxon rank-sum test).

dependency on the texture can be explained by the distinct statistics of luminance and contrast changes when coarse or fine textures move across the RGCs' receptive fields. However, a striking difference existed in suppression recovery times: OFF RGCs on average recovered by ~350 ms after saccade onset, whereas ON RGCs fully recovered only by ~1 s. Similar results were obtained under scotopic conditions for coarse textures, while suppression for fine textures was very weak (Supplementary Fig. 4; all other mouse retina data were recorded at mesopic conditions). In general, the presence of post-saccadic suppression of probe flash responses in both ON and OFF type RGCs suggests a common mechanistic theme across these cell types[4]. On the other hand, the different recovery times indicate either additional suppressive mechanisms in ON RGCs or additional recovery mechanisms in OFF RGCs.

**Spatial origin of retinal saccadic suppression**
*Global component of suppression.* To probe the mechanisms underlying suppression and its differences across ON and OFF type RGCs, we first examined the spatial origin of suppression. We hypothesized that suppression of flash responses was caused by circuits detecting rapid global shifts across the retina. Typically, these circuits include a lateral network of interneurons, communicating with RGCs even from beyond their classical center-surround receptive field (i.e., from their periphery, or far surround)[39,40]. To test whether suppression was caused by such circuits, we modified the spatial layout of the paradigm: we placed a square mask of $1000 \times 1000\ \mu m^2$ (Fig. 2a, right) to restrict the saccades to the periphery of an RGC's receptive field. Similar to the previous experiments, the probe flash was either a dark or bright flash presented over the entire retina, including the masked region. Figure 2b shows the mean population modulation indices of ON RGCs (top) and OFF RGCs (bottom) from these experiments (Supplementary Fig. 5 depicts the underlying population data and shows responses of representative ON and OFF RGCs from these experiments). In OFF RGCs, responses to full-field probe flashes were no longer suppressed when saccades were restricted to the periphery. The responses of ON RGCs, on the other hand, were still suppressed in this condition. The resulting suppression was however weaker and shorter-lived (recovered by 350 ms) than with full-field saccades. These observations (Fig. 2b and Supplementary Fig. 5c) were robust across ON and OFF RGCs whose receptive fields were completely contained within the masked region (Supplementary Fig. 5b).

We will refer to this component of suppression in ON RGCs, which originates from the periphery, as the global component from now on. Such spatially far-reaching inhibition is often mediated through GABAergic wide-field amacrine cells.

We tested this hypothesis by blocking GABA_A receptors. Indeed, in the peripheral saccade condition, the modulation index for most ON RGCs was around 0 in the presence of the GABA_A receptor antagonist SR-95531 (Fig. 2b and Supplementary Fig. 5d). These results suggest that this short-lived global component of suppression is caused by inhibition via GABAergic amacrine cells, perhaps similar to the polyaxonal amacrine cells described previously[20,39,41]. Thus, while suppression is indeed partially caused by circuits detecting global changes across the retina, those circuits seem to act predominantly on ON RGCs, and even there, they only account for a fraction of the total suppression observed with full-field saccades (without mask), which lasts longer. Other, probably more local sources of suppression must exist that account for most of the suppression in ON RGCs and all of the suppression in OFF RGCs.

*Local components of suppression.* To understand the more local components of suppression, we used different analyses and manipulations of the main experimental paradigm. As we will see below, the more local components can be subdivided into a central and a surround component. First, we eliminated the global component, by repeating our normal full-field saccade paradigm in the presence of GABA receptor blockers. The suppression profile of both ON and OFF RGCs was only weakly affected upon blocking GABA_{A,C} receptors (5 μM SR-95531 and 100 μM Picrotoxin; Fig. 2c and Supplementary Fig. 6a). Since the GABA-block eliminates the global component of suppression, the remaining more local components did not seem to rely on GABAergic inhibition. Also, this suggests that the local components dominate retinal saccadic suppression under full-field conditions. We then also blocked glycine receptors (1 μM of Strychnine; Fig. 2d and Supplementary Fig. 6b) to test if the local components of suppression were caused by local inhibition via glycinergic pathways. Here again, the suppression profiles of both ON and OFF RGCs were only weakly affected upon blocking glycine receptors in combination with blocking GABA_{A,C} receptors. Therefore, inhibitory synaptic interactions are not the major mechanism behind the local components, which dominate suppression of RGCs.

Next, we tested whether these local components originated from within the receptive field center. For this, we modified the spatial layout of our paradigm to exclude saccades from the very center of the receptive field. Simply reducing the size of our mask would have severely decreased the number of simultaneously recorded cells located inside the mask, and we therefore resorted to a different strategy: saccades and flashes were presented in small square regions spread across the retina, separated by gaps kept at mean luminance (checkerboard mask, Fig. 2e). In one condition (Fig. 2e, left), we presented saccades and flashes in all

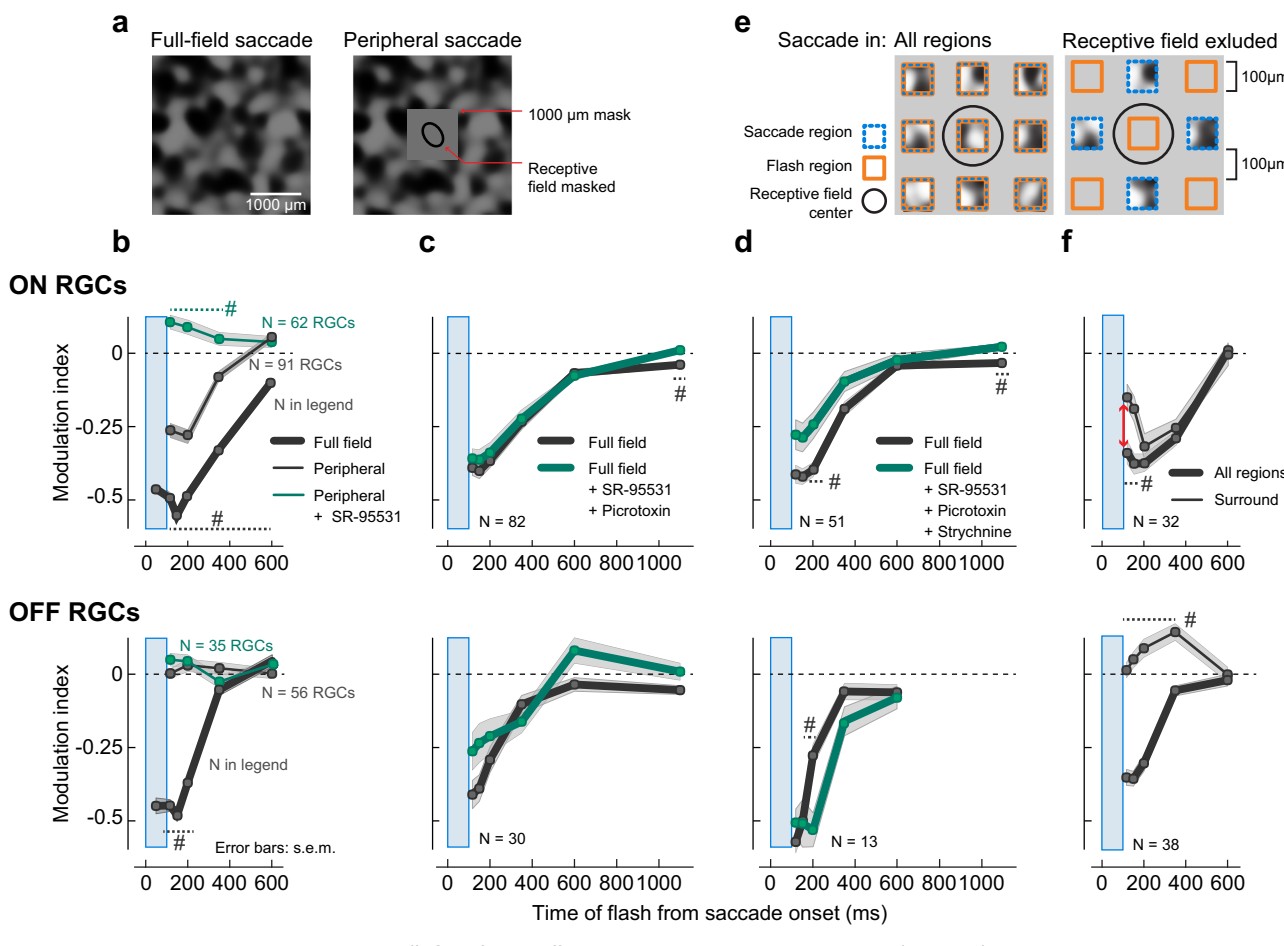

**Fig. 2 Spatial origins of retinal saccadic suppression. a** Spatial layout of the visual stimulation paradigm used in experiments to probe the global component of suppression. Saccades were presented either full-field (left; same as in Fig. 1) or in the periphery (right), where a 1000 × 1000 μm² mask (intensity: mean luminance of texture) covered at least 2-σ of the 2D Gaussian fit to the RGC receptive fields (Supplementary Fig. 5b). **b** Population modulation index (mean ± s.e.m.) of ON (top) and OFF (bottom) RGCs for full-field saccades condition (thick gray lines, same as Fig. 1e rightmost panel; N = 68 to 574 RGCs (see Supplementary Fig. 3 for exact numbers)); periphery saccades condition (thin gray lines; N = 91 ON RGCs, N = 56 OFF RGCs); and periphery saccades condition in the presence of GABA$_A$ receptor blocker (5 μM SR-95531; green lines; N = 62 ON RGCs, N = 35 OFF RGCs). Blue window shows the timing of the saccade. In these experiments, we used a coarse background texture (300 μm spatial scale). Timing of probe flashes: 50 and 150 (only for full-field saccade), 117, 200, 350, 600, and 2100 ms (baseline) after saccade onset. **c, d** Population modulation index (mean ± s.e.m.) of ON (top) and OFF (bottom) RGCs for full-field saccades without any pharmacological agents (gray lines; N = 82 ON RGCs, N = 30 OFF RGCs) and with GABA$_{A,C}$ receptor blockers 5 μM SR-95531 + 100 μM Picrotoxin (**c**; green lines), and for a subset of RGCs where we additionally blocked glycine receptors using 1 μM Strychnine (**d**; green lines; N = 51 ON RGCs, N = 13 OFF RGCs). In these experiments, we used a coarse background texture (150 μm spatial scale). Probe flashes were presented at 117, 150, 200, 350, 600, 1100, and 2100 ms (baseline) after saccade onset. **e** Spatial layout of the visual stimulation paradigm used in experiments to probe local components of suppression. Saccades and flashes were presented in 100 × 100 μm² square regions, separated by 100 μm gaps with mean overall luminance. Left: Saccades and flashes were presented in all regions. Right: Saccades and flashes were presented in alternate regions; only cells with receptive fields (RFs) in the non-saccade regions (orange) were analyzed (black ellipse: 1-σ of the 2D Gaussian fit to an example RGC receptive field). Consequently, saccades were excluded from at most ~300 × 300 μm² of a cell's RF center. In these experiments, we used a coarse background texture (150 μm spatial scale). **f** Population modulation index (mean ± s.e.m.) of ON (top; N = 32) and OFF (bottom; N = 38) RGCs for saccades and flashes in all regions (thick lines) or saccades excluded from RGC RF center (thin lines). Red arrow indicates significant loss in suppression in ON RGCs for early flashes at 117 and 150 ms upon excluding saccades from RF center (P = 0.0016 and P = 0.002, respectively; two-tailed Wilcoxon rank-sum test). In all panels, hash symbols indicate statistically significant difference between groups (P < 0.01, two-tailed Wilcoxon rank-sum test).

regions of the checkerboard mask; in the other condition (Fig. 2e, right), saccades and flashes were presented in alternate regions. With this second arrangement, saccades were excluded from at most ~300 × 300 μm² of a cell's receptive field center, even if that cell was perfectly centered on a non-saccade region. Flashes were presented in the set of regions that included the square region covering the receptive field center of the analyzed RGC (Supplementary Fig. 7a).

Probe flash responses following saccades were suppressed in both ON and OFF RGCs when the saccade and flash were presented in all regions (Fig. 2f, thick lines; Supplementary Fig. 7c; see Supplementary Fig. 7b, e.g., cells), consistent with the suppression observed after full-field saccades (Fig. 1 and Supplementary Fig. 3). When saccades were excluded from the receptive field center, and were presented in alternate regions to the flash, the flash responses were no longer suppressed in OFF

RGCs (Fig. 2f, bottom, thin line), even though these cells showed spiking responses to saccades themselves (Supplementary Fig. 7b). In fact, flash responses were even enhanced. This suggests that the local component of suppression in OFF RGCs arises fully from within the receptive field center (central component). This highly localized origin of suppression in OFF RGCs was further confirmed by additional analysis of the large mask experiments (see Supplementary Fig. 7d, e). In ON RGCs, on the other hand, suppression persisted (Fig. 2f, top and Supplementary Fig. 7c), even though a loss in suppression was apparent for flashes presented immediately after the saccade, at 117 and 150 ms (marked with an arrow in Fig. 2f). This suggests that in ON RGCs, part of the early suppression originates from the central component. The leftover suppression during these early time points might be explained by the global component of suppression, described above (Fig. 2b and Supplementary Fig. 5c), which should also be triggered under this experimental setting. However, since the global component also recovers quickly (by 350 ms, Fig. 2b and Supplementary Fig. 5c), the persisting suppression at the later time points (350 ms and beyond) needs to originate from yet another source beyond the receptive field center. We call this the surround component, and it may originate from the ON RGCs' immediate surround, which also experiences the saccade under this experimental setting. Therefore, in ON RGCs, the local component of suppression can be divided into a central and a surround spatial component.

*Summary of retinal saccadic suppression spatial origins.* In summary, our data suggest that retinal saccadic suppression is mediated by at least three components with distinct spatial origins and temporal properties (Fig. 3): a central, surround, and global component. Suppression in OFF RGCs is mediated exclusively by the central component, which originates from the cell's receptive field center and is characterized by fast onset and

fast recovery (by 350 ms after saccade onset). In ON RGCs, we most directly observed the global component (Fig. 2b and Supplementary Fig. 5a). It extends into the periphery and its timing is similar to the central component in OFF RGCs. Only this global component is affected by blocking GABA receptors (Fig. 2b and Supplementary Fig. 5). During full-field saccades, removing this component by blocking GABA receptors has little effect on the overall suppression (Fig. 2c, d and Supplementary Fig. 6), suggesting a more dominant role of the remaining components. The central component in ON RGCs can only be observed by the loss in suppression for early flashes when saccades are excluded from the receptive field center (marked with an arrow in Fig. 2f, top). Its full duration and time course are obscured by the concurrently acting global and surround components. However, given the identical pharmacological dependencies and spatial origins, it is plausible that the central component is symmetric across ON and OFF RGCs with a common underlying mechanism. Therefore, the longer suppression in ON RGCs can neither be attributed to the central nor global components. It likely originates from the immediate surround of the receptive field. This surround component is long-lasting (recovers by ~1 s) and has a slow onset (Fig. 2f and Supplementary Fig. 7).

**Suppression is triggered by the interaction between consecutive stimuli of opposite polarity.** We previously showed[4] that retinal and perceptual saccadic suppression not only occur after texture displacements, but also after instantaneous texture jumps and structure-free uniform luminance steps. These observations suggested that saccadic suppression is the consequence of rather general mechanisms in which the response to a second stimulus (here: probe flash) gets suppressed by a previous visual transient (caused by saccades or luminance steps). In the following, we apply additional analysis to the luminance-step

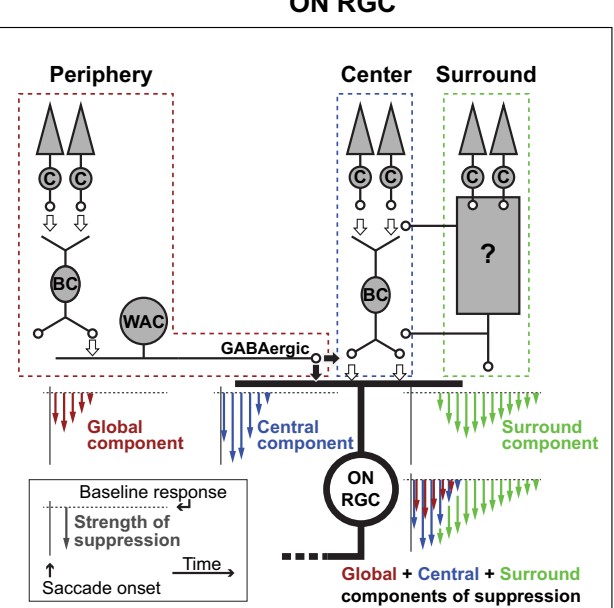
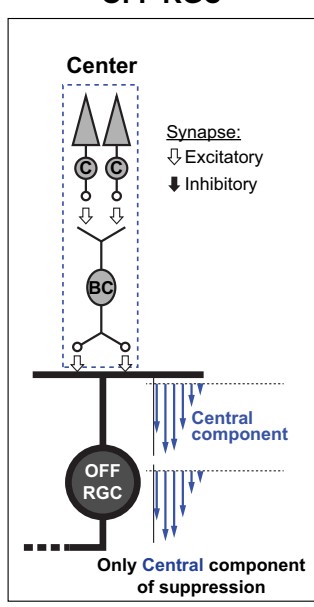

**Fig. 3 Schematic summarizing the spatial origins of retinal saccadic suppression.** Saccadic suppression in OFF RGCs (right) is mediated primarily by the central component of suppression (blue) that originates from the cells' receptive field center. ON RGCs (left) get suppressed from two additional components: First, the fast but short-lived global component (red), mediated by GABAergic inhibition, that originates from as far as the cells' periphery. This global component has a similar temporal profile as the central component. However, it is weaker than the central component and acts in parallel to it, indicated by the red arrows parallel to blue arrows in the total suppression schematic. Second, the delayed but long-lasting surround component (green), which might originate from the cell's immediate surround. The central component and surround component do not depend on classical GABAergic or glycinergic inhibitory pathways. The differences in the suppression recovery time in ON and OFF RGCs was mainly due to this surround component acting on ON RGCs. Inset shows the legend for arrow schematics. Length of the arrows represent suppression strength; spread of the arrows show the temporal profile of suppression.

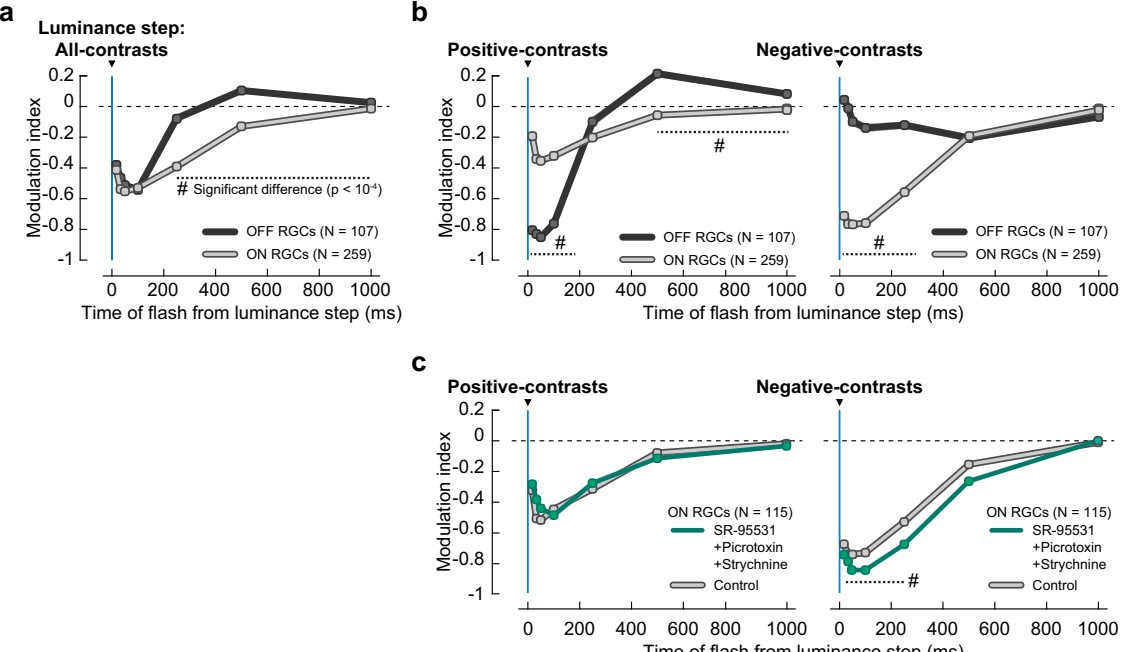

**Fig. 4 Suppression following luminance steps. a** Population modulation index (mean ± s.e.m.) of ON (light gray, $N = 259$) and OFF (dark gray, $N = 107$) RGCs for probe flashes following luminance steps (blue line). Modulation index for each RGC was based on its average response to 56 or 156 luminance-step sequences (Supplementary Fig. 1b) spanning a contrast range of −0.5 to +0.5 Michelson contrast ("Methods"). Probe flashes were presented at 17 ms, 33, 50, 100, 250, 500, 1000, and 2000 (baseline) after luminance steps. Probe flash responses were suppressed in both ON and OFF RGCs, with similar time course and recovery as in the saccade paradigm with textures (Fig. 1e). Error bars are not visible due to small s.e.m. **b** Same as in (**a**), except that the modulation index for each RGC was separately based on average responses to probe flashes after positive-contrast luminance steps (left panel; +0.03 to +0.5 Michelson contrast), and after negative-contrast luminance steps (right panel; −0.03 to −0.5 Michelson contrast). Underlying population data are shown in Supplementary Fig. 8. **c** Same as in (**b**), for a subset of ON RGCs ($N = 115$) in control conditions (light-gray lines) and with $GABA_{A,C}$ and glycine receptors blocked (green lines; cocktail of 5 μM SR-95531, 100 μM Picrotoxin and 1 μM Strychnine). Hash symbol: significant difference between modulation of ON and OFF RGCs in **a** or between ON RGCs without and with pharmacological blockers in (**c**) ($P < 10^{-4}$, two-tailed Wilcoxon rank-sum test).

paradigm (Supplementary Fig. 1b) dataset of ref. [4], to investigate how the polarity and strength of the visual transients affect the suppression of ON and OFF RGCs.

Similar to all previous experiments, we analyzed the modulation index of ON and OFF RGCs separately, using bright probe flashes to analyze ON RGCs and dark probe flashes for OFF RGCs. Consistent with the suppression after texture displacements (Fig. 1e and Supplementary Fig. 3), responses to flashes after luminance steps were strongly suppressed in both ON and OFF RGCs, and ON RGC suppression outlasted suppression in OFF RGCs (Fig. 4a and Supplementary Fig. 8a). The two seemingly different experimental paradigms may therefore trigger similar mechanisms in the retina.

We hypothesized that the response to a luminance step might strongly activate RGCs, so that the response to a subsequent probe flash would drive the cells into adaptation or saturation, effectively resulting in suppressed flash responses. At least for the local components of suppression, this could be a viable mechanism as suppression is not caused by inhibitory synaptic interactions. If this was indeed the case, then positive-contrast luminance steps would suppress responses to bright flashes in ON RGCs, and negative-contrast luminance steps would suppress responses to dark flashes in OFF RGCs. To test this, we separately analyzed the effects of positive- and negative-contrast luminance steps on probe flash responses (Fig. 4b). Surprisingly, the resulting effects were contrary to our adaptation/saturation hypothesis: the responses of ON RGCs to bright probe flashes were only weakly suppressed after positive-contrast luminance steps (Fig. 4b, left), but strongly suppressed following negative-

contrast luminance steps (Fig. 4b, right). Similarly, responses of OFF RGCs to dark probe flashes were weakly suppressed by negative-contrast luminance steps (Fig. 4b, right), but strongly suppressed by positive-contrast luminance steps (Fig. 4b, left). Supplementary Fig. 8 shows the underlying population data for these experiments, and Supplementary Figs. 9a, b shows the spiking response of a representative ON and OFF RGC, respectively. While ON RGCs did show a small component of suppression in support of our adaptation/saturation hypothesis (Fig. 4b left panel, see Supplementary Fig. 9a left column and Supplementary Fig. 10a for a detailed analysis), the dominant suppressive effect in both ON and OFF RGCs was caused by luminance steps with the opposite contrast as the subsequent flash.

Such crossover style of suppression would be consistent with mechanisms involving crossover inhibition via amacrine cells[42], where activation of OFF pathways (here: by the negative-contrast luminance step) would inhibit responses in the ON pathway (here: to the bright probe flash) and vice versa. However, consistent with our earlier experiments (Fig. 2c, d), suppression in ON RGCs still persisted upon blocking $GABA_{A,C}$ and glycine receptors (5 μM SR-95531, 100 μM Picrotoxin, and 1 μM Strychnine) (Fig. 4c). We could not calculate a modulation index for OFF RGCs under these conditions because they did not respond to brief probe flashes in the presence of the pharmacological agents, and therefore the modulation index was mathematically undefined. However, in our texture displacement experiments, the same pharmacological agents (Fig. 2d) had no substantial effect on OFF RGC suppression. The crossover-style

suppression observed in Fig. 4b was therefore unlikely to be caused by classical crossover inhibition pathways involving amacrine cells and GABA_{A,C} or glycine receptors.

### Central component of suppression results from cone response kinetics and nonlinearities in downstream retinal pathways.
Taken together, our experiments so far suggest that suppression in OFF RGCs (1) is mediated solely by the central component of suppression that originates in the receptive field center (Fig. 2e and Supplementary Fig. 7b–e), (2) is predominantly triggered by the interaction between consecutive stimuli with opposite polarity (Fig. 4b and Supplementary Fig. 8b), and (3) is not caused by inhibitory amacrine cells (Fig. 2c, d and Supplementary Fig. 6). Similar conclusions can be drawn for the central component of suppression in ON RGCs. We term this suppressive retinal processing motif which does not rely on inhibition dynamic reversal suppression: dynamic given the required tight temporal link between the two consecutive stimuli; and reversal because the effect is triggered predominantly when the two consecutive stimuli are of opposite contrasts. In this section, we elucidate the mechanism underlying this processing motif.

The highly localized origins of dynamic reversal suppression, and its lack of dependence on inhibition, restrict the possible cellular substrates for this motif to the feed-forward pathway in the retina, namely photoreceptors—bipolar cells—RGCs.

We wondered whether opposite-polarity stimulus–stimulus interactions could already modulate the responses of photoreceptors themselves. For this, we recorded the output of cones with an intensity-based glutamate-sensitive fluorescent reporter (iGluSnFR)[43,44], predominantly expressed in horizontal cells postsynaptic to cone terminals (Methods). We presented a shortened version of the luminance-step paradigm in which a homogeneous background alternated between a brighter and darker gray value (Supplementary Fig. 1c) to induce positive-contrast (+0.4 Michelson contrast) and negative-contrast (−0.4 Michelson contrast) luminance steps. Dark or bright probe flashes (100 ms duration, −0.33 or +0.33 Michelson contrast, respectively) followed the luminance steps at different delays (50, 250, and 2000 ms), with the flash at 2000 ms serving as baseline.

The luminance steps caused sustained changes in the cones' glutamate output (Fig. 5a). The transient responses to the probe flashes were superimposed on these glutamate modulations (Fig. 5b). This superposition was mostly linear and did not indicate nonlinear effects such as adaptation or saturation. Therefore, when we isolated the flash responses by subtracting the step responses, the resulting peak flash responses were only weakly affected by the preceding luminance step (Fig. 5c). Thus, at the level of the cone output (Fig. 5b), there was hardly any suppression when only considering the peak of the probe flash responses (Fig. 5c). How does the suppression observed at the level of RGC output arise from effectively linear cone responses?

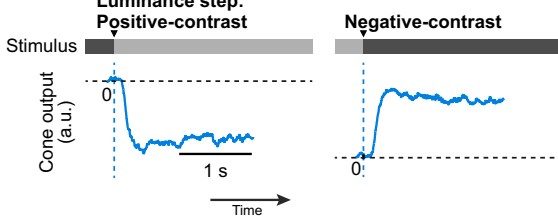

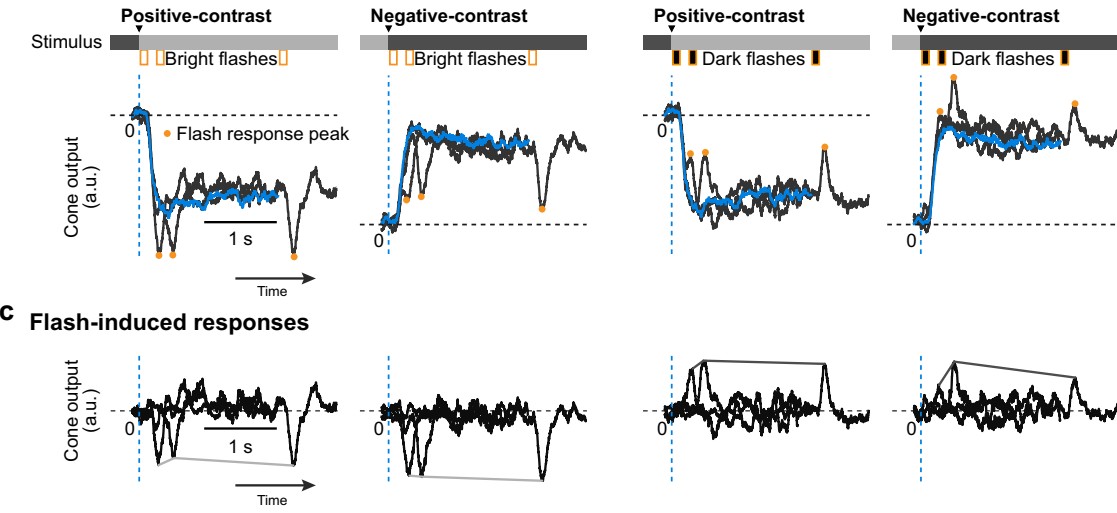

**Fig. 5 Cone output in response to probe flashes following luminance steps. a, b** Cone responses (baseline normalized iGluSnFR indicator fluorescence signal) to positive- and negative-contrast (+0.4 and −0.4 on Michelson scale) luminance steps alone (**a**) and to luminance steps followed by probe flashes at 17, 250, and 2000 ms (orange bars below the intensity bar shows timing of probe flashes) (**b**). Probe flashes were either bright or dark (+0.33 or −0.33 Michelson contrast, respectively; 100 ms long). In (**b**), responses to step-alone (blue) and individual step → flash pairs (dark gray) are overlaid. Dashed blue lines: timing of luminance step; orange circles: peak cone response to flashes; horizontal dashed line: cone level prior to the luminance step. **c** Flash-induced responses, isolated by subtracting luminance step-alone responses (blue) from individual composite luminance-step probe flash responses (dark gray) in (**b**). Lines connecting the response peaks highlight the time courses of suppression relative to baseline flash-induced (2000 ms) responses.

The answer must lie in other properties of the cone response, such as its kinetics, which will be captured by downstream retinal pathways.

To demonstrate this, we used a published computational model[45] that describes RGC spiking responses as a function of the light stimulus. In the model, a light stimulus is passed to model photoreceptors, feeding their output through a set of linear-nonlinear filters which reflect the processing by different bipolar pathways ("Methods"), and eventually converting these filter outputs into RGC spiking. Apart from the model component that captures the cone response and which is described by a differential equation, the model is a simple linear-nonlinear cascade model. Here, as the first step of analysis, we fitted the parameters of the cone component of the model to reflect our measured data of cone output. The model cone faithfully explained the observed cone responses (Supplementary Fig. 11) and gave us the opportunity to calculate cone responses to flashes at additional time points not measured in the experiments. This modeled cone output to step-flash combinations was fed into the model bipolar cells, finally yielding model RGC responses (Fig. 6). In the model, different RGC types can be described by varying the bipolar cell filter properties. We first investigated transient model RGC responses (Fig. 6a–c) and calculated a modulation index (Fig. 6d) comparable with the modulation index of our real RGC data. As a control, we also passed the raw cone output data, instead of the fitted cone model, to the model bipolar cells and found qualitatively the same results (Supplementary Fig. 12).

The model's ON and OFF RGCs (Fig. 6d) showed crossover-style suppression that was consistent with the suppression of real RGCs under similar luminance-step experiments (Fig. 4b): the model's ON RGC showed strong suppression to the bright flashes presented immediately after the negative-contrast luminance step (Fig. 6d, bottom), while bright flashes after the positive-contrast luminance step were only weakly affected (Fig. 6d, top). Suppression recovered by 200 ms, consistent with the recovery time for the central component of suppression in real RGCs. Similarly, the model's OFF RGC showed strong suppression of its response to dark flashes presented immediately after the positive-contrast luminance step (Fig. 6d, top); suppression was absent in OFF RGC when the dark flash was presented later or after a negative-contrast luminance step (Fig. 6d). The markers on the curves in Fig. 6d correspond to the time points when the flashes were presented to the cones in the experiments depicted in Fig. 5. Model RGC responses to step-flash combinations and flash-induced responses at these time points are shown in Fig. 6a–c. In short, the model could replicate dynamic reversal suppression observed in our real RGCs dataset (Fig. 4b).

What properties of the model led to the emergence of the suppressive effect in RGC responses, despite the mostly linear response superposition at the cone output? In the model, the bipolar cells have transient filter properties and are driven predominantly by the instantaneous rate of change in the cone output (i.e., its derivative, rather than by the absolute cone output; Fig. 6e). The response to a probe flash presented immediately after an opposite-contrast luminance step (50 ms) occurred during the initial phase (ramp) of the cone response to the luminance step (Fig. 5b, columns 2 and 3). This causes a much smaller rate of change in cone output than a flash presented after the cone response to a luminance step has already reached its steady-state value (compare flash response at different times in Fig. 5b). This smaller rate of change in cone output therefore drives the downstream bipolar cells only weakly, which then, together with the threshold nonlinearity, results in weak or even completely suppressed model RGC responses (Fig. 6c, columns 2 and 3). In other words, a non-preferred luminance step can

hyperpolarize the retinal pathway for a brief duration. A subsequent flash of preferred contrast presented within that brief duration is therefore less effective because it first needs to reach the threshold. On the other hand, flashes presented during the steady-state phase of the luminance step response (250 ms and 2000 ms in Fig. 5b, columns 2 and 3), or flashes presented immediately after (50 ms) a same-contrast luminance step (Fig. 5b, columns 1 and 4) caused larger instantaneous changes in the cone output, and therefore resulted in relatively stronger spiking (250 ms and 2000 ms in Fig. 6c; 50 ms in Fig. 6c, columns 1 and 4). In conclusion, if two stimuli of opposite contrast occur closely together, an interplay of temporal filtering (emphasizing instantaneous changes) and nonlinear thresholding results in dynamic reversal suppression.

This would suggest that RGCs with different temporal properties (e.g., different transiency) may experience different degrees of dynamic reversal suppression. To test this, we re-analyzed our recorded RGC data of Fig. 4 to quantify suppression as a function of RGC response transiency. We found that suppression did indeed vary with RGC transiency: it was weaker for less transient RGCs, as indicated by the negative slope of the line fit to modulation indices in Fig. 6f; seen in the ON RGC suppression after negative-contrast steps (blue lines in row 2) and OFF RGC suppression after positive-contrast steps (red lines in row 1); columns represent different time points. To explain the origins of this effect, we resorted again to our computational model. Here, a simple change in the temporal filter kinetics could replicate the effect (Fig. 6g): making the filter less transient (Supplementary Fig. 13) led to weaker suppression in the model RGCs. Similarly, the effect could be replicated not by adjusting the linear filter properties, but by making the pathway's thresholding nonlinearity (Eq. (11) in "Methods", Supplementary Fig. 13) stronger (Fig. 6h), consistent with the stricter nonlinearities found in more transient RGC pathways[46].

The analyses in this section suggest that dynamic reversal suppression, the motif underlying the central component of retinal saccadic suppression, does not have a single site of origin. Instead, it appears to emerge from the temporal filtering of the relatively slow kinetics of cone responses (such that flash responses ride on the initial rising/falling phase of the cone's step response), and the subsequent nonlinearities of downstream retinal pathways.

**Generalization to other species**. Retinal saccadic suppression, at least its central component, was triggered by stimulus–stimulus interactions (Figs. 4, 6 and Supplementary Fig. 8), governed by general retinal signal processing, without the need for any specialized saccadic suppression circuit. It is likely that such general processing is conserved across species. Indeed, we observed quantitatively similar retinal saccadic suppression in pig ON and OFF RGCs (Supplementary Fig. 14), including the dependency on background texture statistics. Interestingly, like in mouse, OFF RGCs in pig retina also recovered faster than ON RGCs, suggesting that the surround component of suppression was also present in pig ON RGCs. In an additional experiment, we also recorded the activity of RGCs from an ex vivo macaque retina while subjecting it to a shorter version of the luminance-step paradigm (Supplementary Fig. 1b) with fewer conditions than in the original paradigm. Our results (Fig. 7) indicate that macaque RGC responses to probe flashes, following luminance steps, were suppressed in a way similar to mouse retina. However, more data will be required to determine the population trend and for characterizing the dictionary of response modulations in macaque retina.

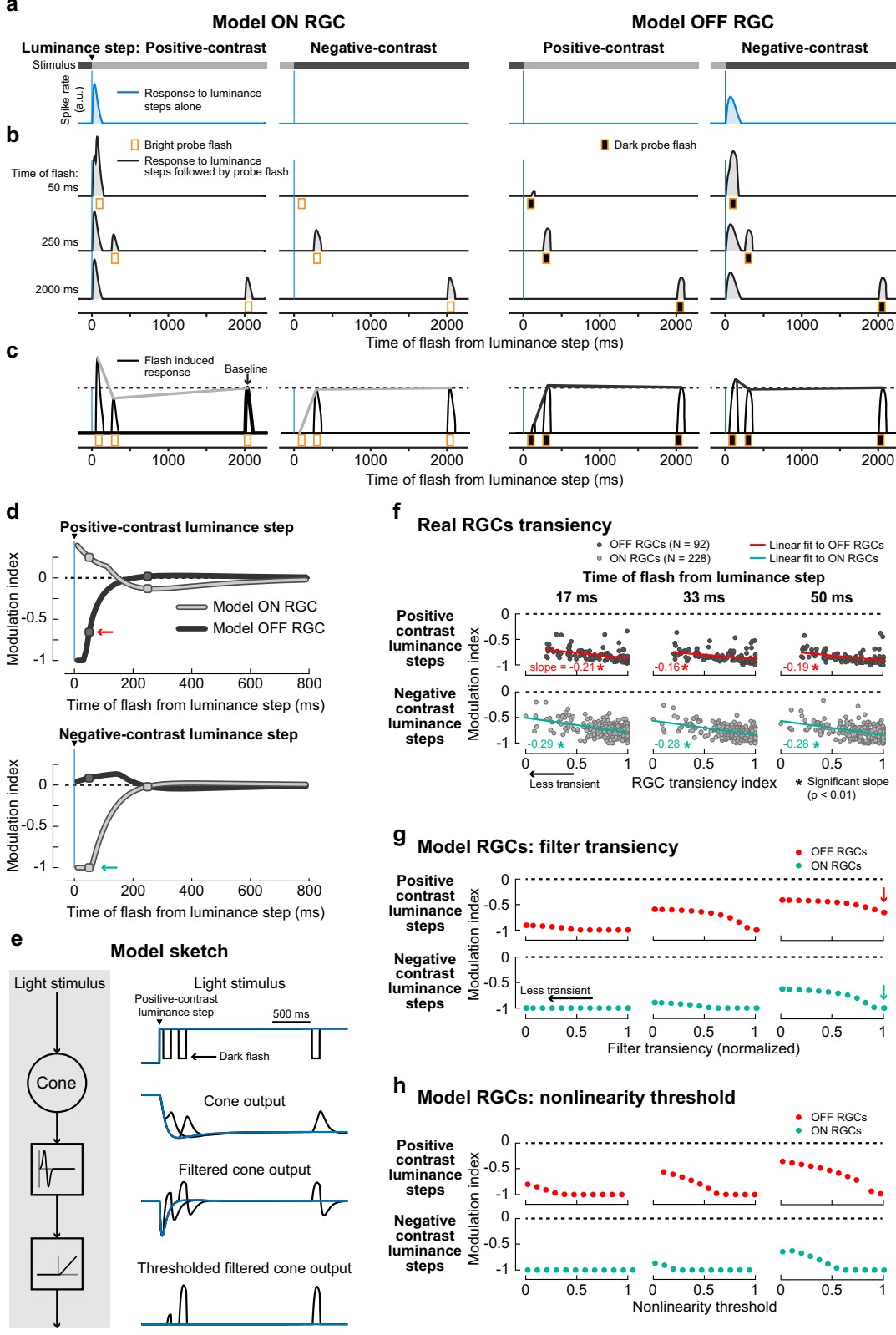

**Downstream visual areas may modulate retinal saccadic suppression**. Given the similarities we previously described between retinal and perceptual saccadic suppression[4], it was tempting to test whether the crossover style of suppression, observed in the retina (Fig. 4b), was also reflected in perception. We therefore conducted human psychophysics experiments where we asked

human subjects ($N = 5$) to maintain saccade-free fixation, while we simply changed the luminance of the homogenous background to a brighter (0.3–0.56 Michelson contrast) or darker (−0.3 to −0.56 Michelson contrast) background (Fig. 8a; "Methods"). At random times relative to the luminance step, we presented a dark (−0.033 Michelson contrast) or bright probe

**Fig. 6 Model RGC responses to probe flashes following luminance steps. a**, **b** Spiking response of model ON (columns 1–2) and OFF (columns 3–4) RGCs to luminance steps alone (**a**, blue) and to luminance steps followed by probe flashes (**b**, black) at 50, 250, and 2000 ms (different rows; analogous to real RGCs in Fig. 1b, c). Luminance steps are depicted by intensity bars in (**a**). First column in each cell type: responses following a positive-contrast luminance step; second column: responses following a negative-contrast luminance step. Vertical blue lines: timing of luminance step; orange bars: timing of probe flashes. Note the ON RGC and OFF RGC did not spike in response to negative-contrast (column 2) and positive-contrast (column 3) luminance steps respectively. **c** Flash-induced responses, after subtracting (**a**) from (**b**), overlaid to show the modulation of probe flash responses at different times (analogous to real RGCs in Fig. 1d). Lines connecting the response peaks highlight the time courses of suppression relative to baseline flash-induced responses (2000 ms). **d** Modulation indices for probe flashes in ON (light gray) and OFF model RGCs (dark gray), following positive-contrast (top panel) and negative-contrast (bottom panel) luminance steps. Modulation indices were calculated based on model responses to probe flashes presented at 10-ms intervals after luminance steps, and baseline as response to a flash at 2000 ms. Circle markers indicate modulation indices based on probe flashes at 50 and 250 ms shown in (**b**, **c**). Cyan and red arrows highlight the suppression of opposite-contrast flashes at 50 ms in ON and OFF RGCs, respectively. **e** Simplified schematic of the model (left), and stimulus and response traces at the different processing steps (right). Signal resulting from light stimulus (in this case positive-contrast luminance step followed by dark flashes) is passed through the cone model and the resulting output is filtered and thresholded. Extended schematic of the model showing different filter transiency and nonlinearity threshold is shown in Supplementary Fig. 13. **f** Modulation indices as a function of RGC transiency for real OFF RGCs (dark gray circles; N = 92; red line: linear regression fit) and ON RGCs (light-gray circles; N = 228; cyan line: linear regression fit). Individual panels correspond to different flash times after positive-contrast (top row) and negative-contrast (bottom row) luminance steps. These RGCs are a subset of the population data shown in Fig. 4b for which we could compute a transiency index ("Methods"). Suppression after negative-contrast steps was weaker in less transient ON RGCs (bottom row, blue regression line has negative slope) and, after positive-contrast steps, suppression was weaker in less transient OFF RGCs (top row, red regression line has negative slope). Numbers in each panel indicate the slope of the fits and asterisk symbol indicates statistically significant slope (slope ≠ 0, P<0.01, two-tailed t test). **g** Modulation index of model OFF (red) and ON (cyan) RGCs as a function of the pathway's transiency, where transiency was varied by changing the transiency of the filter shown in (**e**). **h** Modulation index of model ON and OFF RGCs as in (**g**) but as a function of model's nonlinearity threshold. Arrows in (**g**) highlight the same data as in (**d**). In (**a–d**), the filter transiency was set to 1 and the nonlinearity threshold to 0.1. In (**g**), nonlinearity threshold was set to 0.1. In (**h**), the filter transiency was set to 0. Supplementary Fig. 12 shows model RGC responses based on real cone data of Fig. 5 instead of model cone responses.

flash (+0.033 Michelson contrast), at one of four locations in the subjects' field of view. At trial end, the subjects were asked to localize the probe flash.

Irrespective of the step → flash combination, subjects were strongly impaired in their ability to localize the probe flashes presented around the time of the luminance step (Fig. 8b). Most interesting in this context was the combination of negative-contrast luminance steps with dark probe flashes. In the mouse retina, even though few OFF RGCs did show weak suppression to this combination (Supplementary Fig. 8b, inverted histograms in row 2), this effect was virtually absent at the population level (Fig. 4b, right panel). In human perception, however, this combination led to strong suppression (Fig. 8b, right panel). We cannot exclude that stronger retinal suppression to this specific combination might be present under different light or stimulus conditions. Another possibility is that it might be more pronounced in the retina of humans and other primate species (Fig. 7). Nonetheless, visual mechanisms of suppression certainly exist in higher visual brain areas[4,47]. Perceptual suppression after same-contrast stimulus combinations may arise from processing in these higher visual brain areas, which may modulate and complement retinal saccadic suppression to achieve robust effects at the perceptual level.

**Apparent pre-saccadic suppression in the retina**. Throughout this study we characterized suppression in RGCs of their responses to preferred contrast flashes (bright flashes for ON RGCs and dark flashes for OFF RGCs). Here, we analyze a subset of RGCs in our dataset that also showed responses to flashes of nonpreferred contrast (i.e., some ON RGCs showed responses to dark flashes, and some OFF RGCs to bright flashes). These flash responses were also strongly suppressed around the time of saccade-like image displacements (Fig. 9a, red lines), similar to suppression of preferred contrast flashes (Fig. 9a, green lines), with one key addition: nonpreferred contrast flashes that were presented before saccade onset were also suppressed. This pre-saccadic suppression in the retina is reminiscent of the pre-saccadic suppression observed perceptually[4,17] and neurally in other areas of the brain[10,48]. One explanation for this apparent

pre-saccadic suppression in RGCs is that the responses to non-preferred flashes had higher latencies than responses to preferred flashes. Figure 9b shows responses of an example OFF RGC to preferred and nonpreferred flashes presented 84 ms before saccade onset; Fig. 9c summarizes response latencies for the population of ON and OFF RGCs. Even though the flash occurred before the saccade onset, its peak response did not occur until much later after the saccade offset (Fig. 9b, right; population data shown in Fig. 9c). Suppressive mechanisms triggered by saccade-like image displacements can therefore act on this delayed flash response to suppress it. Measuring modulation of responses as a function of flash time relative to saccade onset (Fig. 9a), like in most studies of saccadic suppression, rather than as a function of response time, then gives the impression of pre-saccadic suppression. We did not further explore if the suppression of responses to nonpreferred stimuli originated from the same central, surround, and peripheral components as identified above (Fig. 3). Nonetheless, it is indeed intriguing that a wide array of observations pertaining to saccadic suppression at the perceptual level, including pre-saccadic suppression, are also observed at the level of the retina.

## Discussion

For most RGCs that we recorded, responses to brief probe flashes were strongly suppressed when presented after saccade-like texture displacements across the retina. Similar suppression occurred when texture displacements were replaced by sudden uniform changes in background luminance, suggesting that suppression was caused by rather generic mechanisms, triggered by visual transients across the retina, rather than specialized suppression circuits that react to image motion. We found that the suppression strength depended on four main factors: (1) strength of the visual transients, governed by the statistics of the background texture or the contrast of the luminance step; (2) elapsed time following the visual transient; (3) RGC polarity (ON vs. OFF RGCs); and (4) RGC response properties (RGC transiency). Stronger visual changes, elicited either by coarser textures or larger luminance-step contrasts, caused stronger suppression, peaking ~50 ms after the stimulus offset (Figs. 1e and 4). The

**a**

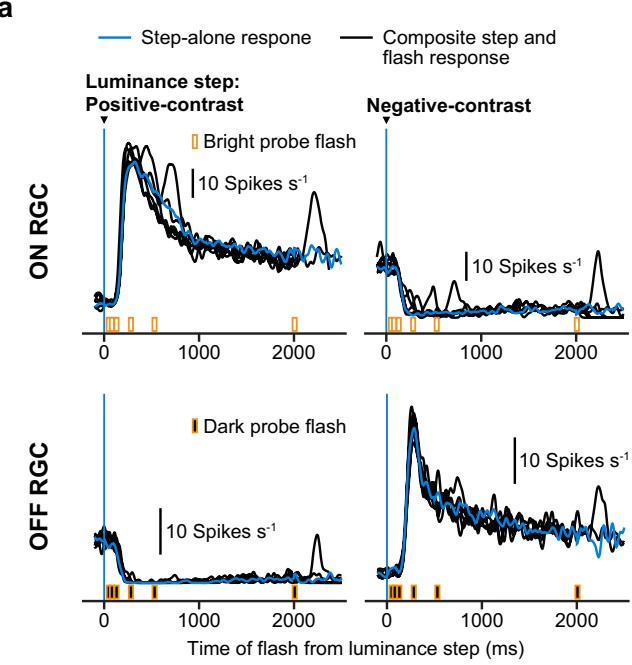

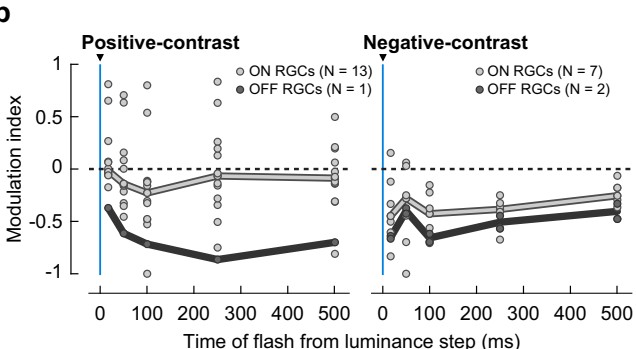

**Fig. 7 Retinal saccadic suppression in Macaque RGCs. a** Average activity of an example macaque ON RGC (top) and OFF RGC (bottom) to positive-contrast (left column) and negative-contrast (right column) luminance steps alone (blue traces) and luminance steps followed by probe flashes (black traces). ON RGCs were analyzed for bright probe flashes and OFF RGCs for dark probe flashes. Responses were averaged across the different positive-contrast (0.05–0.5 Michelson contrast, $N = 10$ sequences) and negative-contrast luminance steps (−0.05 to −0.5 Michelson contrast, $N = 10$ sequences). **b** Median modulation index (thick lines) of macaque ON (light gray) and OFF (dark gray) RGCs for probe flashes presented after positive-contrast (left panel; $N = 13$ ON RGCs, $N = 1$ OFF RGC) and negative-contrast luminance steps (right panel; $N = 7$ ON RGCs, $N = 2$ OFF RGC). Circles represent modulation indices of individual RGCs. This experiment was performed under scotopic light conditions.

recovery times depended on RGC polarity: OFF RGCs recovered by 250–350 ms whereas suppression in ON RGCs lasted for up to 1 s (Figs. 1e and 4). The suppression was stronger in more transient RGCs (Fig. 6f–h). Mechanistically, we identified at least three components of retinal saccadic suppression, with distinct spatial origins, which we defined as central, surround, and global components (Fig. 3). These components were mediated by different underlying mechanisms.

The central component was the only source of suppression we could reliably find in OFF RGCs, and the dominant source in ON RGCs for time points immediately after a full-field saccade or luminance step. This component was short-lived (~250–350 ms),

originated from a cell's receptive field center, and did not depend on inhibitory inputs. It was triggered by opposite-polarity stimulus–stimulus interactions, which naturally occur during saccades and other forms of sequential visual stimulation. This component of suppression resulted from temporal filtering of the relatively slow cone responses to two opposite-polarity consecutive stimuli and the subsequent thresholding nonlinearities. Such a mechanism, where the cone response itself remains linear, but nonetheless forms the basis for subsequent nonlinear response modulation, is clearly different from adaptation[49] or desensitization[50] mechanisms within the cones, which would evoke nonlinear responses of the cones themselves. We call this processing motif, triggered by sequential stimuli of opposite polarity, dynamic reversal suppression. Despite the simplicity of the underlying mechanism, this processing motif substantially shapes the input to downstream visual areas during conditions of natural vision.

The mechanism underlying dynamic reversal suppression also suggests that perceptual saccadic suppression is derived, at least in part, from the inherent response kinetics of photoreceptors, the very first cell in the visual processing cascade. In fact, this early implementation could also explain why we observed suppression in most RGCs we recorded from (Supplementary Figs. 3, 8, and 14), covering a wide spectrum of response properties and therefore presumably many RGC cell types (see Supplementary Fig. 6 of ref. [4]). Still, our results suggest that the suppression initiated at the level of cone photoreceptors is translated differently by the different parallel pathways in the retina, leading to variability in response suppression at the ganglion cell level (such as the stronger suppression in more transient RGCs, Fig. 6f–h). Further cell type classification will be required to relate the degree of modulation with pathway specificity. The type and degree of modulation might also differ across species, even though we see qualitatively similar suppression in mouse, pig, and macaque RGCs.

ON RGCs, in addition to this central component, were suppressed by two more components. First, the global component is a fast but short-lived (~250–350 ms) component, caused by inhibition via GABAergic wide-field amacrine cells, triggered by global image changes and carried to the RGC from as far as the cell's periphery (Fig. 2b and Supplementary Fig. 5). This likely belongs to the same class of circuits that suppresses RGC responses to global motion[20,39]. These circuits were previously suggested to suppress motion awareness during saccades, a phenomenon known as saccadic omission. As indicated by our results, such circuits also contribute towards suppressing RGC sensitivity even after the motion is completed (i.e., saccadic suppression). However, since their influence is masked by more local components of suppression during full-field saccades (Fig. 2c, d and Supplementary Fig. 6), they are unlikely to contribute much to perceptual saccadic suppression. The global suppressive component mediated by GABAergic inhibition is nevertheless one mechanism to process global visual changes, in addition to several others[29]. It may, for example, play a role in perceptual modulations during smooth pursuit eye movements[3]. Here, the central component of suppression will not be triggered in RGCs whose receptive field centers are locked to the tracked object; but these RGCs will still be suppressed by the global component.

The second additional component is the surround component. It seems to act with a delay of ~200 ms and can last for up to 1000 ms. The spatial origins of this component were not investigated in this study, but our data indicate that it presumably arises from the immediate surround of a cell's receptive field. Additionally, similar to the central component, it does not rely on $GABA_{A,C}$ or glycine receptors (Fig. 2c, d and Supplementary

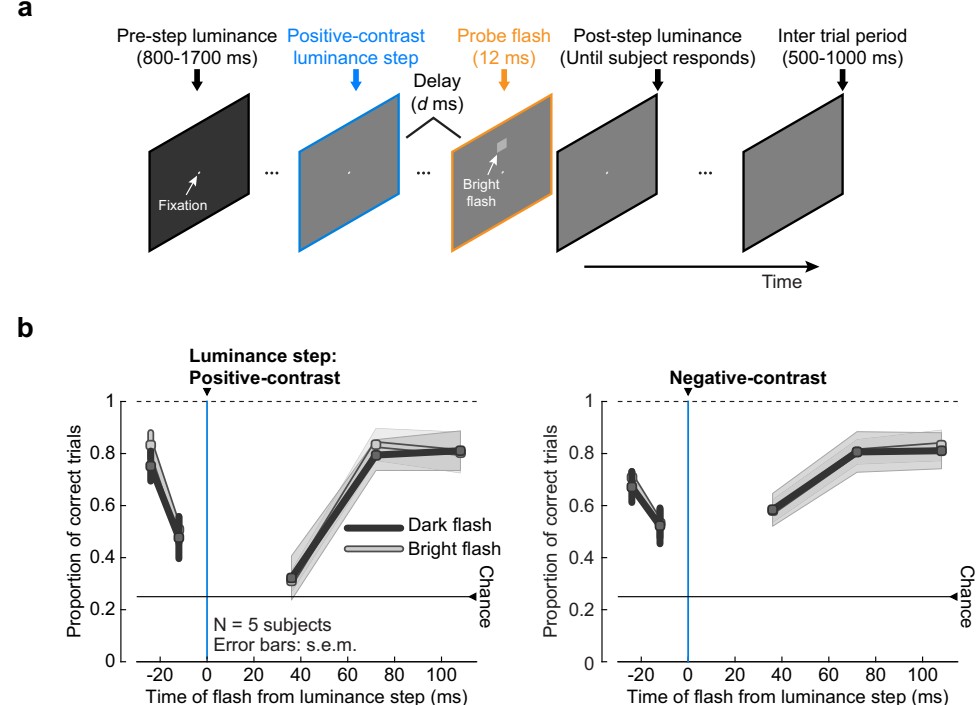

**Fig. 8 Perceptual suppression following luminance steps. a** Example visual task trial. Subjects fixated a small spot on a uniform background with a set luminance (pre-step luminance) for a random duration (800–1700 ms). Background luminance then increased (like shown here) or decreased (positive or negative-contrast luminance step, respectively). At one of five times relative to the luminance step (−24, −12, 36, 72, or 108 ms), a luminance pedestal (probe flash, 147.8 × 147.8 min arc) was applied for ~12 ms at one of four locations relative to the fixation spot: 7 deg above (shown here), below, to the right, or to the left. The probe flash was brighter (shown here) or darker than the current screen luminance. The background remained at the post-step luminance until the subject responded with the perceived location of the flash, plus an additional 500–1000 ms but without the fixation spot, allowing the subject to relax. The current luminance was the pre-step luminance of the consecutive trial. **b** Performance of human subjects (mean ± s.e.m., N = 5 subjects), to correctly localize a dark (dark gray) or a bright (light gray) probe flash presented at different times relative to positive-contrast (left panel) and negative-contrast (right panel) luminance steps (blue line). Each subject's responses were averaged across the different positive-contrast (0.3–0.56 Michelson contrast) and negative-contrast (−0.3 to −0.56 Michelson contrast) luminance steps. Perceptual performance was reduced around the time of luminance steps, reflecting suppression, irrespective of the combination of luminance-step polarity and flash polarity. There were no statistically significant differences in suppression of dark and bright probe flashes (two-tailed Wilcoxon rank-sum test). Note that in the right panel, the suppression profile for bright probe flashes almost completely overlaps the suppression profile of dark probe flashes.

Fig. 6), and the exact mechanisms remain to be explored. Possible mechanisms could involve negative feedback of horizontal cells onto the cones[45,51]. This slower component most likely contributes to the longer recovery times observed in ON cells. Interestingly, visual masking in cat LGN also lasts longer in ON versus OFF cells[52], which may be a consequence of the effects we describe here in the retina. While this surround component plays an important role in shaping RGC and downstream neural responses following a visual transient, its contribution to perceptual saccadic suppression can also be disputed. This is because, during real saccades, eye-movement related signals (e.g., corollary discharge) shorten the duration of suppression caused by visual mechanisms[4], such that the long-lasting surround component may not critically shape perception.

Yet another additional component of suppression, based on saturation-like mechanisms (Fig. 4b left and Supplementary Fig. 10a), was found only in ON RGCs. It is possible that this component originates at the level of bipolar cells, especially because response saturation has been observed predominantly in ON bipolar cells but not in OFF bipolar cells[46]. In summary, while Fig. 3 summarizes the three spatial components of suppression and their temporal properties, these components in turn can have further sub-components.

Given its strength and time course, dynamic reversal suppression is likely the most prominent suppressive retinal

component that contributes towards perceptual saccadic suppression. Yet, irrespective of which components of retinal saccadic suppression contribute towards perceptual saccadic suppression, our results show that retinal responses to stimuli following visual transients are modulated concurrently by several mechanisms (Fig. 3). Additional mechanisms might exist under different stimulus conditions. From the perspective of retinal visual feature processing, this would be consistent with how multiple mechanisms concurrently process other visual features in the retina, such as motion[29].

The retinal suppression that we studied here likely contributes to several other visual phenomena, such as visual masking[53] or neural adaptation with successive stimuli[54]. The similarities between those phenomena and the suppression that we observed in the retina suggest that the retina may be a common neural substrate for these seemingly different types of perceptual phenomena, unifying their underlying mechanisms.

It is remarkable that an elementary property of retinal suppression, i.e., its dependence on the scene statistics (Fig. 1e), is clearly preserved all the way to perception[4]. We also observed pre-saccadic suppression in the retina, but only for responses to flashes with nonpreferred contrast (Fig. 9a). This may, in addition to other mechanisms[55], contribute towards pre-saccadic suppression observed in downstream visual areas[10,48] and perception[4,17]. Perhaps the apparent pre-saccadic suppression in

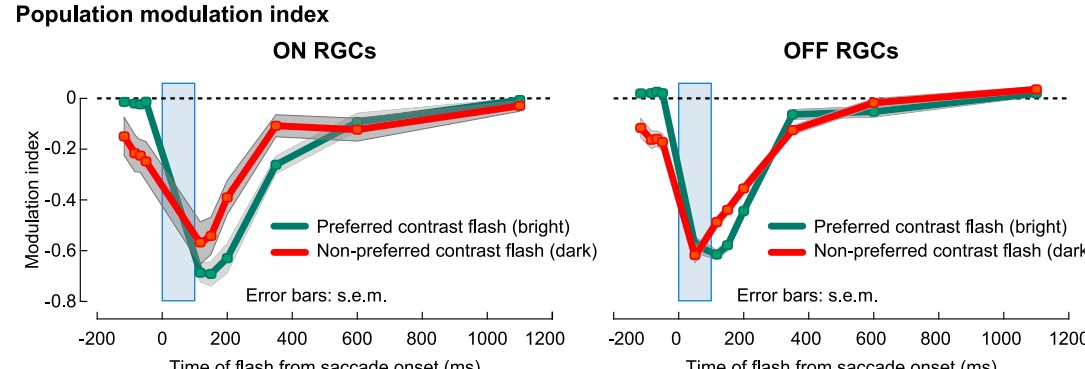

**a** Population modulation index

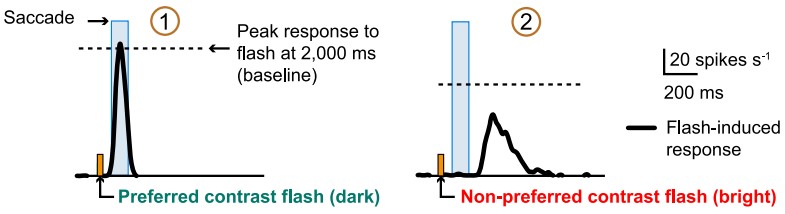

**b** Example OFF RGC response to preferred and non-preferred contrast flashes at -84 ms from saccade onset

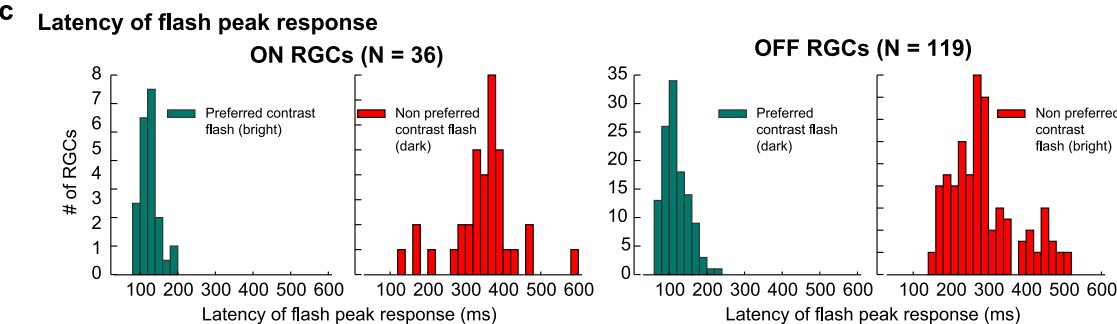

**c** Latency of flash peak response

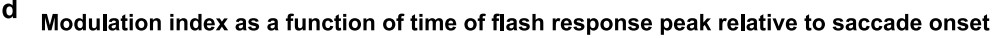

**d** Modulation index as a function of time of flash response peak relative to saccade onset

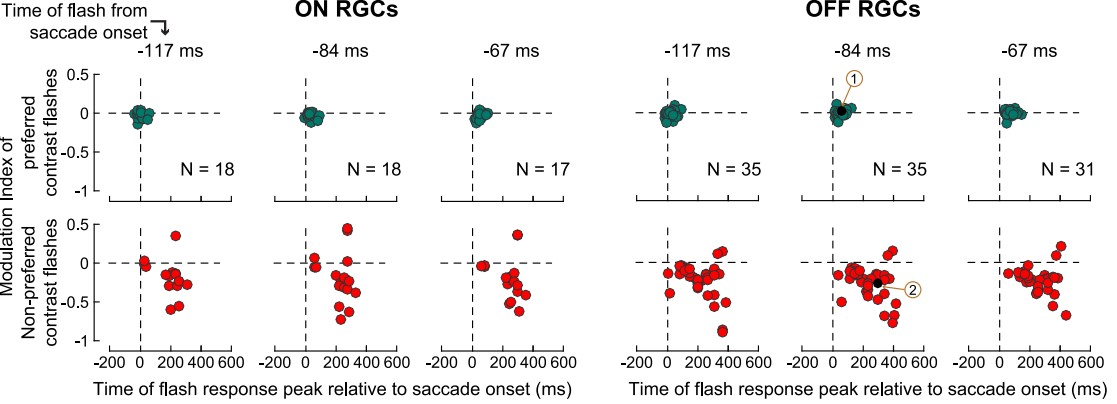

the retina may help elucidate the neural loci and mechanisms of backwards visual masking[53,55,56], a phenomenon observed in downstream visual areas and perceptually, where response to a stimulus is suppressed by a subsequent stimulus. We did not investigate further whether the suppression of nonpreferred contrast flash responses also originates from the components identified in this study. A necessary prerequisite will be to understand the pathways mediating the delayed responses in the case of nonpreferred contrast flashes. Such pathways remain largely unexplored[28].

Not all properties of retinal suppression were preserved in perception. In the retinal output of mice, only successive stimuli of opposite contrasts triggered suppression (crossover-style suppression) (Fig. 4b), while in human perception, all contrast combinations led to strong suppressive effects (Fig. 8). The more far-reaching perceptual suppression may be the result of additional processing beyond the retina[57]. Alternatively, our results may not capture the full array of retinal processing. For one, our stimulus conditions in mouse retina experiments may not have been comprehensive enough to capture all retinal suppressive

**Fig. 9 Suppression of nonpreferred contrast flashes. a** Population modulation index (mean ± s.e.m.) of ON (left) and OFF (right) RGCs for preferred contrast flashes (green lines) and nonpreferred contrast flashes (red lines) presented before and after 150-μm texture- scale displacements (blue bar). For ON RGCs, preferred contrast flashes are bright and nonpreferred contrast flashes are dark. Vice versa for OFF RGCs. Probe flashes were presented at −117, −84, −67, −50, 50, 117, 150, 200, 350, 600, 1100, and 2000 ms (baseline). The number of RGCs in the population varied at each time point; ON RGCs: 18, 18, 17, 17, 0, 17, 17, 17, 36, 17, 29; OFF RGCs: 35, 35, 31, 35, 21, 82, 83, 80, 119, 84, 58. These RGCs are a subset of the population data shown in Fig. 1e (column 3) that also showed robust responses to nonpreferred contrast flashes. Error bars are not visible in the right column due to small s.e.m. **b** Flash-induced response of an example OFF RGC (after subtracting saccade response which is not shown) to preferred contrast (dark) flash (left) and nonpreferred contrast (bright) flash (right) presented 84 ms before saccade onset (indicated by orange markers). Dashed line indicates the baseline (response to flash at 2000 ms after saccade onset) below which a flash response's modulation index is negative. Labels "1" and "2" in brown circles refer to the corresponding points in panel **d**. Response to the preferred contrast flash (left) is not suppressed (peak response close to baseline). Response to nonpreferred contrast flash (right) occurs with a higher latency and is suppressed relative to the baseline. **c** Distribution of response latencies for preferred (green bars) and nonpreferred (red bars) flashes for ON RGCs (left, preferred (mean ± s.e.m.): 117 ± 3 ms, nonpreferred: 347 ± 15 ms) and OFF RGCs (right, preferred: 126 ± 4 ms, nonpreferred: 287 ± 8.2 ms). In both RGCs, nonpreferred contrast flash responses had longer latencies than the preferred contrast flashes. **d** Modulation indices of ON RGCs (columns 1–3) and OFF RGCs (columns 4–6), at different flash times as a function of time of flash response peak relative to saccade onset. Columns: flashes presented at different times before saccade onset. Rows: preferred contrast probe flashes (top) and nonpreferred contrast probe flashes (bottom). Dashed lines correspond to zero modulation on the y axis and saccade onset on x axis. Modulation indices of nonpreferred contrast flashes (red circles) were below 0 even though the flashes were presented before saccade onset (indicated by times above row 1). However, for these nonpreferred flashes, their peak responses occurred long after the saccade onset, as compared to preferred contrast flashes. The longer response latency of nonpreferred flashes therefore gives the impression of pre-saccadic suppression, when suppression is measured as time of flash presentation (like in (**a**)). Black circles in OFF RGC plot for flash at −84 ms are the modulation indices for the two example RGC responses shown in (**b**).

effects. Further, the retinal output of humans and other primate species (Fig. 7) might differ from mouse retina in this respect.

Retinal suppression is only one way that the retina alters its output during dynamic vision. Other forms certainly co-exist, such as brief changes in RGC polarity following peripheral shifts[25] or sensitization of some RGC types following a change in background luminance[19]. These and several other studies[32–35,37–39], together with ours, demonstrate the complex image processing capabilities in the retina to facilitate downstream visual processing for the ultimate service of perception during natural vision. Looking forward, the detailed characterization of retinal output provided here paves the way to investigate the visual features that the retina encodes during dynamic vision. Moreover, it also paves the way to investigate the interactions between retinal and extraretinal (visual and nonvisual) mechanisms of saccadic suppression, to further our understanding of how the visual system maintains stability in the face of constant disruptions.

## Methods

### Experimental model and subject details

*Animals.* We performed electrophysiological experiments on ex vivo mouse, pig, and macaque retinae; and imaging experiments on ex vivo mouse retinae.

Mouse and pig ex vivo retinae experiments were performed in Tübingen, in accordance with German and European regulations, and animal experiments were approved by the Regierungspräsidium Tübingen. Macaque ex vivo retina experiment was performed at Stanford University. Eyes were removed from a terminally anesthetized macaque rhesus monkey used by other laboratories in the course of their experiments, in accordance with the Institutional Animal Care and Use Committee guidelines of Stanford University.

For mouse retina electrophysiology, we used 47 retinae from 15 male and 30 female *PV-Cre x Thy-S-Y* mice (B6;129P2-Pvalb$^{tm1(cre)Arbr}$/J × C57BL/6-tg (ThystopYFPJS)), 3–12 months old, which are functionally wild type[27,28,58]. In addition, we recorded the cone output from four C57BL/6 male mice, 9–10 weeks old. We housed mice on a 12/12 h light/dark cycle, in ambient temperatures between 20 and 22 °C and humidity levels of 40%.

We also replicated experiments on nine pig retinae obtained from six domestic female pigs after they had been sacrificed during independent studies at the Department of Experimental Surgery at the Medical Faculty of the University of Tübingen. Pigs were anesthetized with atropine, azaperone, benzodiazepine (midazolam), and ketamine, and then sacrificed with embutramide (T61). Before embutramide administration, heparin was injected.

One experiment was conducted with a retina extracted from a macaque rhesus monkey.

*Humans.* Human psychophysics experiments were performed in Tübingen. Human subjects provided written, informed consent, and they were paid 10 Euros per session of 60 min each, for three sessions. Human experiments were approved by ethics committees at the Medical Faculty of Tübingen University, and they were in accordance with the Declaration of Helsinki. In total, we collected data from five subjects (24–29 years old; one female).

### Experimental setup: mouse and pig retina electrophysiology.

Mice were dark-adapted for 4–16 h before experiments. We then sacrificed them under dim red light, removed the eyes, and placed eyecups in Ringer solution (in mM: 110 NaCl, 2.5 KCl, 1 CaCl$_2$, 1.6 MgCl$_2$, 10 D-glucose, and 22 NaHCO$_3$) bubbled with 5% CO$_2$ and 95% O$_2$. We removed the retina from the pigment epithelium and sclera while in Ringer solution.

Pigs were dark-adapted for 15–20 min before sacrifice. Immediately after veterinary-confirmed sacrifice, the eyes were enucleated under dim red light, and the cornea, lens, and vitreous were removed. Eyecups were kept in CO$_2$-independent culture medium (Gibco) and protected from light. We transported eyecups to our laboratory and cut pieces from mid-peripheral or peripheral retinae. Only those retinae which showed ganglion cell responses to light stimuli were used in our experiments.

We recorded mouse and pig retinal ganglion cell (RGC) activity using either low- or high-density multi-electrode arrays (MEAs). The low-density setup consisted of a perforated 60-electrode MEA (60pMEA200/30ir-Ti-gt, Multichannel Systems (MCS), Reutlingen, Germany) having a square grid arrangement and 200 μm inter-electrode distance. We whole mounted an isolated retina on a nitrocellulose filter (Millipore) with a central 2 × 2 mm hole. The mounted retina was placed with the RGC side down into the recording chamber, and good electrode contact was achieved by negative pressure through the MEA perforation. We superfused the tissue with Ringer solution at 30–34 °C during recordings, and we recorded extracellular activity at 25 kHz using a USB-MEA-system (USB-MEA1060, Multichannel Systems) or a memory-card based system (MEA1060, Multichannel Systems). Data was acquired using MC Rack version 4.6.2 (Multichannel Systems). A detailed step-by-step approach is provided in ref. [59].

The high-density MEA setup consisted of either a HiDens CMOS MEA[60] (developed by the lab of Andreas Hierlemann, Basel, Switzerland) or a MaxOne system[61] (Maxwell Biosystems, Basel, Switzerland). The HiDens CMOS MEA featured 11,011 metal electrodes with inter-electrode (center-to-center) spacing of 18 μm placed in a honeycomb pattern over an area of 2 × 1.75 mm. Any combination of 126 electrodes could be selected for simultaneous recording. The MaxOne MEA featured 26,400 metal electrodes with center-to-center spacing of 17.5 μm in a grid-like arrangement over an area of 3.85 × 2.1 mm. In this system, up to 1024 electrodes could be selected for simultaneous recordings. For each experiment, a piece of isolated retina covering almost the entire electrode array was cut and placed RGC-side down in the recording chamber. We achieved good electrode contact by applying pressure on the photoreceptor side of the retina by carefully lowering a transparent permeable membrane (Corning Transwell polyester membrane, 10 μm thick, 0.4-μm pore diameter) with the aid of a micromanipulator. The membrane was drilled with 200 μm holes, with center–center distance of 400 μm, to improve access of the Ringer solution to the retina. We recorded extracellular activity at 20 kHz using FPGA signal processing hardware. In the case of the HiDens CMOS MEA, data were acquired using custom data acquisition software, called MEA 1k Scope (developed by the lab of Andreas Hierlemann, Basel, Switzerland). In the case of the MaxOne MEA, data were

acquired using MaxLab software provided by Maxwell Biosystems, Basel, Switzerland.

In total, we performed 59 recordings, 47 from mouse and 12 from pig retinae. In total, 24 of the 59 recordings were done using low-density MEAs. Once a basic experimental protocol was established, we shifted to HiDens CMOS MEA providing much higher throughput. 12 experiments were done using this setup. We upgraded to the MaxOne MEA for even higher throughput and did 23 recordings using this setup. A subset of the data collected from 32 of the 59 recordings (20 from mouse and 12 from pig retinae), was also used in our previous study[4]. Here, we show further in-depth analysis of that data.

We presented light stimuli to the retinal piece that was placed on the MEA using a DLP projector running at 60 Hz (Acer K11 for low-density MEA experiments and Lightcrafter 4500 from EKB Technologies Ltd. with internal red, green and blue light-emitting diodes, for high-density MEA experiments). 60 Hz is above the flicker fusion frequency of both mouse and pig retinae; therefore, the framerate of these projectors was adequate for our purposes. The Acer K11 projector had a resolution of $800 \times 600$ pixels covering $3 \times 2.25$ mm on the retinal surface. Lightcrafter 4500 had a resolution of $1280 \times 800$ pixels, extending $3.072 \times 1.92$ mm on the retinal surface. We focused images onto the photoreceptors using a condenser (low-density MEA recordings, illumination from below) or a ×5 objective (high-density MEAs, illumination from above). In each case, the light path contained a shutter and two motorized filter wheels with a set of neutral density (ND) filters (Thorlabs NE10B-A to NE50B-A), having optical densities from 1 (ND1) to 5 (ND5). The filters allowed us to adjust the absolute light level of the stimulation.

We measured the spectral intensity profile (in $\mu W\ cm^{-2}\ nm^{-1}$) of our light stimuli with a calibrated USB2000 + spectrophotometer (Ocean Optics) and converted the physical intensity into a biological equivalent of photoisomerizations per rod photoreceptor per second ($R^*rod^{-1}\ s^{-1}$), as described in ref. [28]. Light intensities of the projector output covered a range of 3 log units (i.e., 1000-fold difference between black and white pixels, over an 8-bit range). We linearized the projector output, and we used only grayscale images of limited contrast, spanning at most the range from 0 to 120 in the 8-bit range of the projector (see stimulus description below for details). Absolute light intensities were set to the mesopic level, where a stimulus intensity of "30" in our 8-bit DLP projector scale (0–255) corresponded to 225–1000 $R^*rod^{-1}\ s^{-1}$, depending on the experimental rig used for the experiment (i.e., different DLP projectors and MEAs). We pooled all data from the different rigs as separate individual analyses from the individual setups revealed no effects of recording conditions in the different setups. For experiments of Supplementary Fig. 4, we also recorded at scotopic light levels where a stimulus intensity of "30", corresponded to 23 $R^*rod^{-1}\ s^{-1}$ at the scotopic level.

**Experimental setup: macaque retina electrophysiology.** In one experiment, we recorded the activity of macaque retinal ganglion cells. For this experiment, we used a high-density MEA[62,63]. Following enucleation, the anterior portion of the eye and vitreous were removed. The eye was stored in a dark container in oxygenated Ames' solution (Sigma, St. Louis, MO) at 33 °C, pH 7.4. Under infrared illumination, a small piece of retina ~$1 \times 1$ mm, from a retinal region with eccentricity around 12 mm (4.0–17 mm temporal equivalent eccentricity; Chichilnisky and Kalmar[64]), was dissected and placed ganglion cell side down on a MEA for recording. The retina pigment epithelium remained attached during the recording; the retina was perfused with oxygenated Ames' solution. A custom planar large-scale MEA[63,65] with a hexagonal outline of 519 electrodes at 30 μm pitch was used. Recorded voltages were band-pass filtered, amplified, and digitized at 20 kHz using custom electronics[65]. The detailed spike sorting process is described in ref. [63].

Visual stimulation was performed with the optically reduced image of a gamma-corrected OLED microdisplay (eMagin) refreshing at 60.35 Hz focused on the photoreceptor outer segments. The visual stimulus was delivered through the mostly-transparent electrode array. The power of each display primary was measured at the preparation with a calibrated photodiode (UDT Instruments). At the mean background illumination level, the photoisomerization rates for the rods and the L, M, and S cones were ~29, 9, 9, and 2 $P^*receptor^{-1}\ s^{-1}$, respectively (see ref. [66]), placing the retina in a scotopic regime.

**Experimental setup: cone photoreceptor imaging.** To record the output of cone photoreceptors in the mouse retina, we measured the glutamate release using an intensity-based glutamate-sensitive fluorescent reporter, iGluSnFR[43] expressed in horizontal cell processes postsynaptic to cone terminals, using a viral approach. We recorded the cone output from 4 retinae obtained from two C57BL/6 male mice, 9–10 weeks old. Below, we reproduce the methods, previously described in ref. [44].

We dark-adapted the mice for ≥1 h before the experiments. They were then anaesthetized using isoflurane (Baxter) and sacrificed by cervical dislocation. The eyes were enucleated and hemisected in carboxygenated (95% $O_2$ and 5% $CO_2$) artificial cerebrospinal fluid (ACSF) solution containing (in mM): 125 NaCl, 2.5 KCl, 2 $CaCl_2$, 1 $MgCl_2$, 1.25 $NaH_2PO_4$, 26 $NaHCO_3$, 20 glucose, and 0.5 L-glutamine (pH 7.4). We then moved the tissue to the recording chamber where it was continuously perfused with carboxygenated ACSF at ~36 °C. In these experiments, ACSF contained ~0.1 μM Sulforhodamine-101 (SR101, Invitrogen) to reveal blood vessels and any damaged cells in the red fluorescence channel[67]. All procedures were carried out under very dim red (>650 nm) light.

iGluSnFR was expressed in the retina by viral transduction of AAV2.7m8.hSyn.iGluSnFR, generated in the Dalkara lab (Institut de la Vision) as described in refs. [68,69]. The iGluSnFR plasmid construct was provided by J. Marvin and L. Looger (Janelia Research Campus, USA). A volume of 1 μL of the viral construct was injected into the vitreous humor of the mice, anaesthetized with 10% Ketamine (Bela-Pharm GmbH & Co. KG) and 2% xylazine (Rompun, Bayer Vital GmbH) in 0.9% NaCl (Fresenius). For the injections, we used a micromanipulator (World Precision Instruments) and a Hamilton injection system (syringe: 7634-01, needles: 207434, point style 3, length 51 mm, Hamilton Messtechnik GmbH). Imaging experiments were performed 3–4 weeks after injection. In the outer retina, iGluSnFR was predominantly expressed in horizontal cells. As the expression tended to be weaker in the central retina, most scan fields were acquired in the medial to peripheral ventral or dorsal retina.

To record the iGluSnFR signal, we used a MOM-type two-photon microscope setup (designed by W. Denk, MPI, Heidelberg; purchased from Sutter Instruments/Science Products). The design and procedures are detailed in refs. [67,70]. In brief, the system was equipped with a mode-locked Ti:Sapphire laser (MaiTai-HP DeepSee, Newport Spectra-Physics), two fluorescence detection channels for iGluSnFR (HQ 510/84, AHF/Chroma) and SR101 (HQ 630/60, AHF), and a water immersion objective (W Plan-Apochromat ×20/1.0 DIC M27, Zeiss). The laser was tuned to 927 nm for imaging iGluSnFR. For image acquisition, we used custom-made software (ScanM by M. Müller and T. Euler) running under IGOR Pro 6.3 for Windows (Wavemetrics), taking time-lapsed $128 \times 128$ pixel image scans at 3.9 Hz in the outer plexiform layer (OPL).

For light stimulation in cone imaging experiments, we used the Lightcrafter (LCr; DPM-E4500UVBGMKII), a DLP projector from EKB Technologies Ltd. with internal UV and green light-emitting diodes (LEDs). The light from the DLP projector was focused through the objective. To optimize the spectral separation of mouse M- and S- opsins, LEDs were band-pass filtered (390/576 Dualband, F59-003, AHF/Chroma). LEDs of the DLP projector were synchronized with the microscope's scan retrace. Stimulus intensity (as isomerization rate, $P^*cone^{-1}\ s^{-1}$) was calibrated to range from ~500 (black image) to ~20,000 for M- and S-opsins. In addition, a steady illumination component of ~$10^4\ P^*cone^{-1}\ s^{-1}$ was present during the recordings because of two-photon excitation of photopigments. The overall light intensity falling onto the retina was therefore in the low photopic regime. The light stimulus was centered to the recording field before every experiment. For all experiments, the retinal tissue was kept at a constant mean stimulator intensity level for at least 15 s after the laser scanning started and before stimuli were presented.

**Experimental setup: human psychophysics.** Subjects sat in a dark room 57 cm in front of a CRT monitor (85 Hz refresh rate; 41 pixels per deg resolution) spanning $34.1 \times 25.6$ deg (horizontal × vertical). Head fixation was achieved with a custom head, forehead, and chin rest[71], and we tracked eye movements of the left eye at 1 kHz using a video-based eye tracker (EyeLink 1000, SR Research Ltd, Canada). Gray backgrounds in the luminance-step experiment (Fig. 8) were always presented at an average luminance of 49.84 $cd\ m^{-2}$, and the monitor was linearized (8-bit resolution) such that equal luminance increments and decrements for luminance steps were possible around this average. In total, we collected data from five subjects (24–29 years old; one female). A subset of the data from four subjects was used in our previous study[4]. Here, we perform further analyses of the complete dataset, in addition to one new subject.

**Pharmacology.** In several MEA experiments, we used pharmacological agents to block specific receptors in the mouse retina. To block $GABA_A$ receptors selectively, we used 5 μM SR-95531 (gabazine, an antagonist of $GABA_A$ receptors; Sigma). To block both $GABA_A$ and $GABA_C$ receptors, we used 100 μM picrotoxin (an antagonist of $GABA_A$ and $GABA_C$ receptors; Sigma). To block glycine receptors, we used 1 μM Strychnine (antagonist of Glycine receptors).

We first prepared a 1000× stock solution of these pharmacological blockers as follows: SR-95531 was dissolved in water at a concentration of 5 mM; picrotoxin was dissolved in DMSO at a concentration of 100 mM; Strychnine was dissolved in Chloroform at a concentration of 1 mM. During the experiments, we pipetted the stock solution to the Ringer solution in a 1:1000 ratio. Wash-in was performed for 20 min.

**Visual stimuli for the saccade paradigm used in retina electrophysiology (Figs. 1 and 2 and Supplementary Figs. 3–7, 14).** In retina electrophysiology experiments, we used two broad visual stimulation paradigms: a saccade (texture displacements) paradigm (Supplementary Fig. 1a), and a luminance-step paradigm (Supplementary Fig. 1b), described in detail below. In different experiments, we used different spatial and/or pharmacological manipulations of these two paradigms.

*Background textures.* We created background textures (Supplementary Fig. 2a) by convolving a random binary (i.e., white or black) pixel image with a two-

dimensional Gaussian blurring filter[72] defined by the kernel

$$G(x, y) = e^{\frac{-(x^2+y^2)}{2\sigma^2}} \qquad (1)$$

The parameter $\sigma$ of the kernel influenced the amount of blurring. This resulted in textures having effectively low-pass spectral content (Supplementary Fig. 2b) with a cutoff frequency depending on $\sigma$. For easier interpretation, we define the spectral content of these textures by a spatial scale. Intuitively, the spatial scale approximates the size of the smallest dark and bright image blobs of the texture (Supplementary Fig. 2a). Quantitatively, the spatial scale is defined as the $2*\sigma$ parameter of the Gaussian blurring filter. We generated textures with four different spatial scales: 25, 50, 150, and 300 μm, that resulted in dark and bright image blobs approximating a range of receptive field sizes between bipolar cells (texture with spatial scale 25 μm, see ref. [73]) and RGCs (textures with spatial scale 150 and 300 μm). In other words, coarser textures matched the resolution of RGCs, and finer textures matched the resolution of one processing stage earlier, the retinal bipolar cells. Calculating power spectra for the textures (Supplementary Fig. 2b) confirmed that the spatial scale and hence the cutoff frequencies were consistent with this design aim. In different experiments, we used textures of all or a subset of the different spatial scales.

We normalized the pixel intensities in the textures to have uniform variations in luminance around a given mean. We used pixel intensities (from our 8-bit resolution scale) ranging from 0 to 60 around a mean of 30, or ranging from 30 to 90 around a mean of 60 (see sub-section "Saccades and probe flashes" for when each paradigm was used).

*Saccades and probe flashes.* To simulate saccades in our ex vivo retina electrophysiology experiments, we displaced the texture across the retina in 6 display frames (100 ms at 60 Hz refresh rate). For easier readability, we usually refer to these saccade-like texture displacements as saccades. The textures were displaced in each frame by a constant distance along a linear trajectory. While each saccade lasted 100 ms, displacement direction was varied randomly for each saccade (uniformly distributed across all possible directions), and saccade amplitude could range from 310 μm to 930 μm (corresponding to a velocity range of 3100–9300 μm s$^{-1}$ on the retinal surface). In visual degrees, this corresponds to a velocity range of 100–300 deg s$^{-1}$ and displacement range of 10–30 deg in mice, well in the range of observed mouse saccade amplitudes[74]. Similar to primates, mice also have oculomotor behavior, even under cortical control[75]. For example, they make, on average, 7.5 saccade-like rapid eye movements per minute when their head is fixed[74] (humans make several saccades per second). We used the same retinal displacement range of 310 μm to 930 μm for pig retinae. To the best of our knowledge, pig oculomotor behavior has not been documented in the literature. However, with their larger eyeball sizes, our translations of the retinal image would correspond to slower saccades (e.g., small saccades in humans and monkeys), which are also associated with saccadic suppression. Moreover, retinal saccadic suppression is not critically dependent on the details of movement kinematics, as it is triggered by visual transients (Fig. 4, also see Figs. 4, 5 in ref. [4]).

Each trial consisted of successive sequences (Fig. 1a and Supplementary Fig. 1a) that combined a saccade with a probe flash, as follows: there was first a pre-saccade fixation of 2 s, where the texture remained static over the retina, then a 100 ms saccade, followed by post-saccade fixation where the texture again remained static over the retina but now with a shifted texture. At a certain time from saccade onset (delay $d$, range: 50 ms to 2100 ms), we presented a probe flash (see below). Following the probe flash, the texture remained static at the post-saccade fixation position for another 2 s before the next saccade of the successive sequence occurred. The post-probe flash fixation of one sequence was therefore also the pre-saccade fixation of the next sequence. This way the texture remained visible during the entire trial, being translated during saccades of the successive sequences. In a single trial, 39 such sequences occurred. In each successive sequence, the direction and amplitude of the saccade were pseudo-randomly determined by the range of allowed saccade amplitudes and directions. The texture always landed at unique locations within a trial. The end result was that, within a single trial, RGCs experienced a wide spectrum of saccade amplitudes, directions, and contrasts across these 39 saccades. As such, by analyzing the average effects of the 39 saccades on RGC responses to probe flashes, we captured a wide range of saccade-induced kinematics and luminance changes over the RGC receptive fields.

In most cases, the probe flash had a duration of 2 frames (~33 ms). We used 1 frame (~16 ms) in a subset of earlier experiments (mouse: 709 of 1616 cells; pig: 116 of 228 cells). Results were pooled across these paradigms as the effects were indistinguishable. The probe flash was a full-screen positive (bright) or negative (dark) stimulus transient.

Bright or dark probe flashes could happen in two different ways across our experiments. The results were indistinguishable between the two ways, so we pooled results across them. Briefly, in one manipulation, the probe flash replaced the texture with a homogeneous bright (pixel intensity of 60 in our 8-bit projectors) or dark (pixel intensity of 0) full-screen (in these experiments, the textures themselves had intensities ranging from 0 to 60 pixel intensity; see "Background textures" above). This way, the flash contrast from the underlying background luminance was variable across space (e.g., a bright flash on a bright portion of a texture had lower contrast from the underlying texture than the same flash over a dark portion of the texture). In the second manipulation, the bright and dark

flashes were simply luminance increments or decrements (by pixel values of 30 on our 8-bit projectors) over the existing textures. This way, spatial contrast relationships in the background textures were maintained. In these experiments, the textures themselves had a range of 30–90 pixel intensities and a mean pixel value of 60 (on our 8-bit projectors). Out of the 1616 RGCs that we analyzed for saccadic suppression across all experiments where texture displacements were used as saccades (irrespective of the spatial or pharmacological manipulations), 1129 RGCs experienced such probe flashes, whereas the rest (487 RGCs) experienced the homogenous probe flash. For pig retina recordings, we always used the homogenous framework. However, in the subset of pig experiments where the 2-frame probe flash was employed (112 of 228 RGCs), we used a high-contrast probe flash such that a bright flash would be achieved by first going to 0 in the first frame of the flash and then going to 60 (on our 8-bit projectors) in the next frame (and vice versa for a dark flash). Again, all data were pooled across these different paradigms because their outcomes were indistinguishable.

The number of trials required during a physiology experiment depended on the number of conditions that we ran on a specific day. For example, testing 7 different flash delays required 15 trials (7 with bright probe flashes, 7 with dark probe flashes, and 1 without probes). In a given experiment, we always interleaved all the conditions; that is, in any one of the 15 necessary trials, each of the 39 saccades could be followed by a bright or a dark probe at any of the 7 delays, or no probe flash at all (Supplementary Fig. 1a shows schematic of one such trial). Moreover, we repeated the total number of conditions (e.g., the interleaved 15 trials) four times per session, and we averaged responses across those repetitions. Since one trial typically lasted for 2 min, the example of 15 trials repeated 4 times lasted for ~2 h. This was usually combined with additional conditions (e.g., other background texture scales). Therefore, the total number of saccades shown in any given experiment could be computed by #trials × 39 saccades per trial × #textures × #repetitions. A typical experiment lasted 10–12 h. If the combination of conditions would have required even longer recordings in a given experiment, we reduced the number of conditions (e.g., we presented flashes at fewer delays, or used fewer texture scales).

*Full-field saccades.* In the full-field saccades experiments, saccades and probe flashes occurred over the entire retina. This was the main experimental paradigm that we used for characterizing how saccades modulate RGC responses to probe flashes. This paradigm was also used as a control in experiments in which we applied different spatial or pharmacological manipulations of this paradigm to probe for the spatial origins and mechanisms of saccadic suppression. Further, results from this paradigm served as a baseline standard across different experimental rigs. This paradigm was used in a total of 32 retinal recordings (32 retinae from 30 mice). In different recordings, we used a subset of the texture scales and probe flash delays. This explains the different values of N seen for different conditions in, for example, Fig. 1e and Supplementary Fig. 3. However, to ensure comparison, some conditions always overlapped across different recordings. This paradigm was also used in 12 recordings with retinae from 6 pigs.

*Periphery saccades (global component of suppression).* In this manipulation, we restricted saccades to the RGC's receptive fields periphery (i.e., its far surround). This spatial manipulation was used to investigate the spatial origins of the global component of suppression (Fig. 2a, b and Supplementary Fig. 5). We performed 13 recordings (13 retinae from 13 mice) with this paradigm, always with the high-density multielectrode array system (MaxOne by MaxWell), as it provided a large electrode area (~2 × 4 mm$^2$) for the retina to be placed on. The recording region was typically either a high-density block of electrodes (inter-electrode spacing: ~17.5 μm) or a block with one-electrode spacing (inter-electrode spacing: ~35 μm). The recording region was selected close to the center of the electrode array. We centered a large square mask (1000 × 1000 μm$^2$) over the recording region to restrict the texture and saccade presentation to the periphery of RGC receptive fields (Supplementary Fig. 5b). The mask had a homogeneous intensity corresponding to the mean luminance of the texture. At different times relative to texture displacements, full-field probe flashes were presented, similar to experiments with full-field texture displacements. The intensity of each pixel of the stimulus (both the mask and the texture regions) was adjusted for the probe flashes, either decreased or increased by a pixel value of 30 (on our 8-bit projectors) for dark and bright probe flashes, respectively. In all periphery saccade experiments, we used a probe flash duration of ~33 ms, and a coarse texture background of spatial scale 300 μm.

*Checkerboard mask paradigm (local component of suppression).* In this spatial manipulation, we presented saccades and flashes in small square regions spread equidistantly over the entire retina. Each square region measured 100 × 100 μm$^2$, separated from adjacent squares by an edge-edge gap of 100 μm. The gap was kept at mean luminance throughout the experiment. Saccades and flashes could either be presented in all the regions (similar to full-field saccades, except for the gap), or in alternate regions (Fig. 2e and Supplementary Fig. 7a), arranged like in a checkerboard. This paradigm was used to investigate the origins of the local component of suppression (Fig. 2f and Supplementary Fig. 7b, c). We performed four recordings (four retinae from three mice) with this paradigm, always with the low-density MCS MEA rig. In all experiments with this spatial manipulation,

we used probe flash duration of ~17 ms, and a coarse texture background of spatial scale of 150 μm.

**Visual stimuli for luminance-step paradigm used in retina electrophysiology (Figs. 4 and 7 and Supplementary Figs. 8–10).** In this paradigm (Supplementary Fig. 1b), we used no textures at all. The screen was always a homogenous gray field, and the visual event of a saccade was replaced by an instantaneous step to a different gray value. The gray backgrounds had intensities between 30 and 90 (on our 8-bit projector). The instantaneous step in intensity caused either a positive-contrast luminance step (in the range of +0.03 to +0.50 Michelson contrast) or a negative-contrast luminance step (−0.03 to −0.50 Michelson contrast). This paradigm was used to characterize the stimulus-stimulus interactions that ultimately trigger retinal saccadic suppression (Figs. 4, 7 and Supplementary Figs. 8–10). We performed a total of four recordings (four retinae from four mice) with this paradigm, always with the high-density MaxOne MEA rig. A trial consisted of either 56 or 156 successive sequences (Supplementary Fig. 1b) that each combined a luminance step with a probe flash, as follows: there was first a pre-step fixation of 2 s where the retina was exposed to a fixed gray level (analogous to pre-saccade fixation in texture displacements), then an instantaneous switch to post-step fixation (analogous to post-saccade). At a certain time from the luminance step (delay: 17, 33, 50, 100, 250, 500, 1000, or 2000 ms), we presented a 2-frame (~33 ms) dark (−0.33 Michelson contrast) or bright (+0.33 Michelson contrast) probe flash. Some sequences contained no probe flash, the next luminance step then happened 4 s after the previous one. In a given experiment, we had 17 trials representing the 17 conditions: 8 flash delays × 2 probe flash polarities +1 condition with no probe flash. Similar to the saccade paradigm, we always interleaved all conditions; that is, in any one of the 17 necessary trials, each luminance step could be followed by a bright or a dark probe at any of the 8 delays, or no probe flash. Moreover, we repeated the 17 trials at least four times.

A shorter version of this paradigm was used in our macaque retina recording (Fig. 7). Here, a trial consisted of 20 successive sequences. The 20 luminance steps induced contrasts in the range −0.5 to +0.5. Flashes of ~33 ms were presented with a delay of 17, 50, 100, 250, 500, and 2000 ms after the luminance step.

**Visual stimuli for RGC characterization batch used in all retina electrophysiology experiments.** We used other stimuli unrelated to the main experiments to help us characterize RGC properties (e.g., response polarity, latency, transiency, and spatial receptive fields). These stimuli had the same mean intensities and intensity ranges as the textures or luminance steps used in each experiment. Below, we describe these stimuli for the condition in which the texture intensities ranged from 0 to 60 pixel intensity (represented as grayscale RGB values in the units of our 8-bit projector). In experiments in which the textures ranged in intensity from 30 to 90, or the luminance-step experiment, all intensities reported below were shifted upward by 30. (1) Full-field contrast steps. ON steps: stepping from 0 to 30 (+1 Michelson contrast) and from 30 to 60 (+0.33) for 2 s. OFF steps: stepping from 60 to 30 (−0.33) and from 30 to 0 (−1) for 2 s. (2) Full-field Gaussian flicker, 1 min. Screen brightness was updated every frame and was drawn from a Gaussian distribution with mean 30 and standard deviation 9. This stimulus was used to calculate linear filters representing the temporal receptive fields of RGCs through reverse correlation (spike-triggered averaging of the stimulus history). (3) Binary checkerboard flicker, 10–15 min. The screen was divided into a checkerboard pattern; each checker either covered an area of 55 × 55 μm, 60 × 60 μm, or 65 × 65 μm depending on the recording rig. The intensity of each checker was updated independently from the other checkers and randomly switched between 10 and 50 or 0 and 120. This stimulus also allowed us to calculate linear filters representing the spatial receptive fields of RGCs.

**Visual stimuli for cone photoreceptors imaging (Figs. 5 and 6).** For cone imaging experiments, we used a minimalistic version of the luminance-step paradigm used in retina electrophysiology. A homogeneous background alternated between a darker (pixel intensity 50 on 8-bit projector) and brighter gray value (pixel intensity 120 on 8-bit projector); the transitions between these two background values represented positive and negative contrast of 0.4 Michelson contrast. At various times after the luminance step (50, 250, and 2000 ms) we presented a probe flash (100 ms duration, +0.33 or −0.33 Michelson contrast). The probe flash at 2000 ms served as the baseline. The next background transition always happened 2 s after the preceding probe flash. The combination of the two luminance steps and the two probe flash polarities yielded a total of four combinations: negative-contrast luminance step followed by dark flash; negative-contrast luminance step followed by bright flash; positive-contrast luminance step followed by dark flash; and positive-contrast luminance step followed by a bright flash. A single trial (Supplementary Fig. 1c) was composed of the four step-flash combinations occurring three times (for the three delays with which the flash was presented); and the negative- and positive-contrast luminance step without a flash. Within a trial, these conditions were randomized. A trial was repeated three times. The luminance steps and the flashes were presented within a 700-μm disc region centered over the scan field.

**Visual stimuli for human psychophysics (Fig. 8).** In the human psychophysics experiment (Fig. 8), we mimicked the retinal luminance-step experiments of Fig. 4. The paradigm (Fig. 8a) was similar to the one described in[4]. Subjects fixated a central fixation spot over a gray background that remained there for the entire duration of a trial. The background had one of 8 luminances (22.4, 30.24, 38.08, 45.92, 53.76, 61.6, 69.44, 77.28 cd m⁻²). After a random initial fixation duration (800–1700 ms after fixation spot onset), the luminance of the background was changed suddenly (in one display frame update) to one of the remaining 7 luminances, inducing a positive-contrast luminance step or a negative-contrast luminance step. In our analysis, we used the luminance steps that induced contrasts in the range +0.3 to +0.56 Michelson contrast and −0.3 to −0.56 Michelson contrast. At one of five different possible times relative to the time of the luminance step (−24, −12, 36, 72, or 108 ms), a luminance pedestal (probe flash) was applied briefly for one display frame (~12 ms) at one of four locations relative to display center (7 deg above, below, to the right of, or to the left of center). Note that because the display was rasterized (that is, drawn by the computer graphics board from the top left corner in rows of pixels), the exact flash time and duration depended on the location of the flash on the display (but in a manner like other psychophysical experiments studying the same phenomenon, and also in a manner that is unlikely to affect our results). The luminance pedestal consisted of a square of 147.8 × 147.8 min arc, in which we added or subtracted a value to represent bright and dark probe flashes. We ensured that the contrast of the flash (relative to the currently displayed background luminance) was always the same across all trials: +0.033 for a bright flash, and −0.033 for a dark flash. Following each trial, the fixation spot was removed from the background to allow the subjects to relax. This inter-trial period lasted for 500–1000 ms. The next trial happened consecutively, in a way that the current luminance of the background was used as the pre-step luminance. Subjects maintained fixation throughout all trials (except the inter-trial period) and simply reported the locations of the brief flashes. Each subject performed three sessions, with 1120 trials per session.

**Data analysis for retina electrophysiology**

*MEA recordings preprocessing.* Low-density MEA recordings were high-pass filtered at a 500 Hz cutoff frequency using a tenth-order Butterworth filter. We extracted spike waveforms and times using thresholding, and we semi-manually sorted spikes using custom software. For high-density MEA recordings, we performed spike sorting by an offline automatic algorithm[76] and assessed the sorted units using a custom-developed tool, the UnitBrowser[77]. We judged the quality of all units using inter-spike intervals and spike shape variation. Low-quality units, such as ones with high inter-spike intervals, missing spikes, or contamination, were discarded. All firing rate analyses were based on spike times of individual units. In total, we extracted 3510 high-quality units after the spike sorting (referred to as RGCs from now on), from recordings of mouse retina. From pig retina recordings, we extracted 376 RGCs and from macaque retina, we extracted 57 RGCs after the spike sorting. However, as we mention below, only a subset of these could be analyzed for saccadic suppression.

*RGCs characterization using receptive fields, ON–OFF index, transiency index.* We first characterized the properties of RGCs. We calculated linear filters in response to full-field Gaussian flicker and binary checkerboard flicker by summing the 500-ms stimulus history before each spike. The linear filters allowed for determining cell polarity. Specifically, the amplitude of the first peak of the filter was used: If the peak was positively deflected, the cell was categorized as an ON cell; if negatively deflected, the cell was an OFF cell. ON cells were later always analyzed with respect to their responses to bright probe flashes, and OFF cells were analyzed with dark probe flashes. We determined the spatial receptive fields of RGCs by calculating the linear filters for each region (checker) defined by the binary checkerboard flickering stimulus. The modulation strength of each linear filter, measured as the standard deviation (s.d.) along the 500 ms temporal kernel, is an estimate of how strongly that region drives ganglion cell responses. We fitted the resulting 2D-map of s.d. values with a two-dimensional Gaussian and took the 2-σ ellipse (long axis) as the receptive field diameter. For all other figures and analyses, we converted spike times to estimates of firing rate by convolving these times with a Gaussian of σ = 10 ms standard deviation and amplitude 0.25 σ⁻¹e^{1/2}.

For each RGC, we used responses to full-field contrast steps to calculate an ON–OFF index, a transiency index, and a response latency index. These indices were used to characterize the properties of RGCs that we included in our analyses. The ON–OFF index was calculated by dividing the difference between ON and OFF step peak response by their sum. The resulting index values ranged between −1 (OFF) and +1 (ON) and were then scaled to span between 0 (OFF) and +1 (ON). The transiency index was defined as the ratio of the response area within the first 400 ms and the total response area spanning 2000 ms. The resulting index had a value of 1 for pure transient cells. Response latency was calculated as the time from stimulus onset to 90% of peak response. This value was normalized to the maximum response latency in our dataset to create the response latency index.

*Modulation index.* To quantify retinal saccadic suppression, we first determined a baseline response, defined as the response to a probe flash ~2 s after texture displacement onset or 2 s after luminance step (delay between 1967 and 2100 ms, depending on the specific flash times used in a specific experiment). This baseline

response was compared to responses of the same cell to the same flash when it occurred at an earlier time (i.e., closer in time to the saccade). Usually, the saccade-like texture displacements themselves caused significant neural responses (saccade-alone response, e.g., Fig. 1b), and the responses to the flashes were superimposed on these saccade-responses. We therefore first isolated the component of the responses caused by the flashes by subtracting the saccade-alone responses from the composite saccade and flash responses. We refer to this isolated component as the flash-induced responses.

To get a robust estimate of the response to saccades-alone (i.e., without any flashes), we averaged spike rate from before saccade onset up until the next saccade onset for conditions in which no flash was presented, or until just before the flash onset for conditions in which a post-saccade flash was presented. This was done for each of the 39 successive saccades in a given trial.

We then computed a neural modulation index, ranging from $-1$ to $+1$. A value of $-1$ represents complete suppression of flash-induced responses, whereas $+1$ indicates complete enhancement of flash-induced responses (that is, there was only a response to a flash after saccades, but not to a flash in isolation). A modulation index of 0 meant no change in flash-induced response relative to the baseline response. The modulation index of an RGC for a given flash of preferred contrast (bright flash in ON RGCs and dark flash in OFF RGCs) delay $d$ after saccade onset was calculated as $(r_d - r_b)/(r_d + r_b)$ where $r_d$ is the peak firing rate for the flash-component of the response (see above for how we isolated this from the composite saccade+flash response) and $r_b$ is the peak firing rate for the baseline flash response (i.e., the same flash but occurring ~2 s away from any saccade; see above). Here, peak firing rate was taken as the maximum firing rate within 300 ms time window after the flash onset of the averaged response from all repetitions of a given condition (delay $d$ or baseline) for a given RGC, across all saccades.

The calculation of the modulation index of an RGC for a given flash of nonpreferred contrast (dark flash in ON RGCs and bright flash in OFF RGCs; Fig. 9) differed from the above procedure. This is because firstly, response latencies varied greatly across RGCs, and secondly for some cells, the peaks were not well defined. Therefore we used a template matching method similar to the one described in ref. [78]. Briefly, for a given RGC, we first extracted its baseline flash response and used that as a template, including the timing of this response relative to flash onset. Then, for each flash delay $d$ after saccade onset, we subtracted the saccade-only response from the response to the saccade-flash combination to isolate the flash-induced component of the response, which we then compared to the template. We quantified the strength of the flash-induced response component (relative to baseline) by making a least-squared-error linear fit of the template to that response component as follows:

$$response \sim s * template \qquad (2)$$

The linear term $s$ of this fit was then interpreted as the relative response strength, with a value <1 indicating a weaker response relative to the baseline flash. The modulation index was then calculated as $(s_d - 1)/(s_d + 1)$ where $s_d$ is the relative response strength of flash delay $d$ from saccade onset. For preferred contrast flashes, this method and the one described in the previous paragraph produced similar modulation index values.

To quantify the modulation at a population level, we averaged the modulation indices of the individual RGCs in that population. For some analyses, we also calculated modulation indices of RGCs for each of the 39 individual saccades using the same procedure.

In some cells, individual saccades from the sequence of 39 were discarded. For example, imagine that "saccade No 3" gets discarded. This would happen when the baseline response strength (response to the probe flash with 2 s delay) after saccade No 3 is weak (specifically: peak amplitude less than 60% of the median of all 39 baseline response strengths). We did this to ensure that our modulation indices were not marred by a denominator approaching zero (e.g., if both flash and baseline responses were weak). We did, however, re-include some sequences. For example, if the probe flash with delay 100 ms after saccade No 3 triggered a strong response (specifically: peak amplitude larger than the median baseline response peak across the 39 saccades), then saccade No 3 would be re-included for the condition "100 ms delay". This was done in order to re-include sequences (if discarded by the first step) for which the baseline flash response was weak but a flash after saccades nonetheless gave a robust response. For example, this could happen if a cell did not respond to the baseline flash but the saccade enhanced the response to a flash following it.

Since the modulation index was based on responses to the brief probe flashes, it could only be computed for RGCs that did respond to these brief flash stimuli. In our analysis, we included all such RGCs. Of the spike sorted RGCs across all paradigms, we included: 2002 of 3510 in mice (47 retinae); 228 of 376 in pigs (pieces from 12 retinae); and 15 of 57 in macaque (1 retina).

**Full-field saccades.** We analyzed 1010 mouse RGCs (633 ON; 377 OFF) and 228 pig RGCs (197 ON; 31 OFF) for saccadic suppression using the full-field version of the saccade paradigm. A subset of data from 688 of the 1010 mouse RGCs and a subset of data from all the 228 pig RGCs was presented previously[4]. Here, we perform further analyses on the complete datasets from the RGCs recorded previously, in addition to analyzing the newly recorded RGCs. For each RGC, we quantified the modulation index for a full-field probe flash presented at different times from saccade onset.

A subset of these RGCs were also tested for saccadic suppression while blocking GABAergic and glycinergic inhibition. For 82 ON and 30 OFF RGCs, we had a direct comparison with and without $GABA_{A,C}$ blockers (5 μM SR-95531 + 100 μM Picrotoxin) (Fig. 2c and Supplementary Fig. 6a). For 51 ON and 13 OFF RGCs, we had a direct comparison with and without $GABA_{A,C}$ blockers in addition to glycine blocker (1 μM Strychnine; Fig. 2d and Supplementary Fig. 6b).

In yet another subset of RGCs (72 ON; 49 OFF), we also analyzed saccadic suppression at the scotopic light level (Supplementary Fig. 4), which was 1 log unit dimmer than the light level at which all other recordings were performed. For these cells, we had a direct comparison of responses at scotopic and mesopic levels.

**Periphery saccades (global component of suppression).** For each recorded RGC, we computed a masking factor (post hoc) to quantify how well its receptive field was covered by the $1000 \times 1000\ \mu m^2$ mask. We first determined the spatial receptive field center of each RGC (described above in "RGCs characterization"). The masking factor was defined as the multiple of $\sigma$ of the two-dimensional Gaussian fit for which the ellipse just touched the mask boundary (Supplementary Fig. 7d). Cells with receptive field centers within the mask were defined to have a positive masking factor while those lying outside were given a negative masking factor. The magnitude of the factor increased with distance from the edge of the mask. This way, cells for which the mask covered their receptive field centers and immediate surround had masking factors >2 (these were the cells included in the analysis shown in Fig. 2b and Supplementary Fig. 5); cells with a mask covering only the receptive field center had masking factors between 1 and 1.5. Cells located close to the mask's edge, with masking factors between $-1$ to $+1$, had their receptive field centers partially exposed to the saccade. Finally, cells lying outside the mask where the receptive field center always experienced saccades had masking factors < $-1$. A total of 642 RGCs (401 ON; 241 OFF) from 13 experiments were recorded with this spatial layout of the saccade paradigm. Cells for which clear receptive fields could not be calculated were excluded from any further analysis. The exact number of cells for different conditions within this paradigm are reported in the results section. For each RGC, we calculated the modulation index for flashes presented at different times from saccade onset, in the same manner as described under "Modulation index". Supplementary Fig. 7e shows the modulation index of individual RGCs as a function of its masking factor. Only a subset of RGCs with masking factors in the range $-3$ to $+5$ were included in this analysis. The median modulation index was calculated by taking the median of modulation indices of RGCs within a 1.2 masking factor window, sampled at intervals of 0.1 (running median).

In a subset of these experiments, we analyzed the effects of blocking GABAergic inhibition (using 5 μM SR-95531) on the modulation of probe flash responses in RGCs with masking factors >2, i.e., with receptive field centers and immediate surround effectively masked. This included 62 ON and 35 OFF RGCs with robust responses to brief probe flashes with the pharmacological agent; only these RGCs were analyzed for saccadic suppression under the presence of SR-95531 (Fig. 2b). In a subset of these RGCs (29 ON; 25 OFF), we had a direct comparison of modulation indices with the control condition, i.e., periphery saccades in the absence of SR-95531 (Supplementary Fig. 5d).

**Checkerboard mask regions saccades.** In experiments where we investigated the local component of suppression (Fig. 2e, f and Supplementary Fig. 7b, c), we had three stimulus presentation settings (Supplementary Fig. 7a): (1) saccade and flash presented in all squares regions; (2, 3) saccades and flashes in alternate regions of a hypothetical binary checkerboard. For each recorded RGC we calculated the modulation index for the following two scenarios: saccades and flashes in all regions (Supplementary Fig. 7a presentation setting 1); and saccades excluded from the receptive field center (Supplementary Fig. 7a presentation setting 1 or 2 depending on the RGCs spatial position relative to the presentation area). For this, we first calculated the spatial receptive field center (see heading "RGCs characterization"; receptive field center was defined as the 1-$\sigma$ ellipse of the 2D Gaussian fit) for each RGC. We then calculated the Euclidean distance between the receptive field center and the closest flash region for both presentation settings 2 and 3. For further analysis, we used the presentation setting with the shortest distance between a flash region and the receptive field center. In case this flash region is perfectly centered over the receptive field center, saccades will be excluded from a region of at most 300 μm diameter centered over the receptive field center. In most cases, the RGCs were indeed well centered within a flash region since the center of each region coincided with electrodes in the low-density MEA. Nonetheless, for each RGC we calculated the intersection between its receptive field area and saccade regions in pixels (where each pixel corresponded to 3.75 μm on the retinal surface). RGCs for which more than 15% of their receptive field area intersected with the saccade regions were excluded from further analysis. In the end, a total of 51 RGCs (32 ON; 38 OFF) from 4 retinae recordings were used for further analysis of saccadic suppression.

**Luminance-step paradigm.** To quantify retinal saccadic suppression with the luminance-step paradigm (Figs. 4 and 7), we used the same analyses and statistical procedures to those described above for the saccade (texture displacement) paradigm. The only difference was that instead of 39 successive sequences in a trial, we now had either 56 or 156 successive sequences (or 20 in case of macaque retina

experiment), spanning a contrast range of ±0.03 to ±0.5 Michelson contrast. Similar to the texture displacement paradigms, the modulation index was based on responses to brief probe flashes (~33 ms flash duration), and it could therefore only be computed for cells that did respond to these flash stimuli ($N = 366$ of 668 spike sorted RGCs from four mouse retinae; $N = 15$ of 57 spike sorted RGCs from a macaque retina). The modulation index for ON RGCs ($N = 259$ mouse RGCs; $N = 13$ macaque RGCs) was calculated from responses to bright probe flashes, and that for OFF RGCs ($N = 107$ mouse RGCs; $N = 2$ macaque RGCs) was calculated from responses to dark flashes. A subset of data from all mouse RGCs was presented previously[4]. Here, we perform further analyses on the complete dataset of the same RGCs.

In a subset of luminance-step experiments (two mouse retinae), we analyzed the effects of blocking GABAergic and glycinergic inhibition on the modulation of probe flash responses. In total, 115 ON RGCs showed robust responses to the brief probe flashes with and without the pharmacological blockers. However, none of the OFF RGCs responded robustly to the baseline probe flash in the presence of pharmacological blockers and therefore OFF RGCs were excluded in the quantification of saccadic suppression in the presence of pharmacological blockers.

We also analyzed if the modulation of flash-induced responses depended on the strength of the response to the preceding luminance step. This analysis was done to establish whether suppression of flash responses resulted from saccade-induced saturation or adaptation of ganglion cell responses (Supplementary Fig. 10). For each RGC, we calculated an association index that quantified the monotonic relationship between response to the luminance step and response to subsequent flashes. We first binned the responses across the 56 or 156 step-flash sequences (Supplementary Fig. 1b) based on the contrast induced by the luminance step in each sequence. Bin width was set to 0.025 Michelson contrast. Then, within each bin, we averaged the responses to luminance steps alone (Supplementary Fig. 9a, b, top row) and to luminance steps followed by probe flashes. For each probe flash delay and contrast bin, we quantified the strength of the response induced by the luminance step preceding each probe flash (we integrated the average response to the luminance steps followed by probe flashes, up until the response to that probe flash). The modulation index was calculated as usual. We then calculated Spearman's rank correlation coefficient (R) between the modulation index and the response strength induced by the luminance step, across all the contrast bins. This can be visualized from the insets in Supplementary Fig. 9. For ease in the interpretation of the results, we termed the resulting correlation coefficient as the association index. Intuitively, this association index describes the monotonic relationship between the step response strength and the strength of suppression and can be interpreted as follows: the larger the magnitude of the association index, the stronger is the monotonic relation between the two quantities. A negative value indicates that stronger step responses are associated with decreasing (more negative) modulation indices (i.e., weaker flash responses → suppression). A positive value indicates that stronger step responses are associated with increasing modulation indices (i.e., stronger flash responses → less suppression or even enhancement). In the example cell of Supplementary Fig. 9a, the association index has large negative values for flashes immediately after the positive luminance step, suggesting that a stronger step response is indeed strongly correlated with stronger suppression of subsequent flashes. A robust calculation of association index was only possible for luminance steps that activated the RGC (i.e., positive-contrast luminance steps in ON RGCs, negative-contrast luminance steps in OFF RGCs). Supplementary Fig. 10 shows the association index for each RGC and flash time as a function of the cell's modulation index.

In Fig. 6f, we plot modulation index as a function of RGC transiency index (see "RGCs characterization" for details on transiency index). The RGCs shown in this sub-figure were a subset of the RGCs analyzed with the luminance-step paradigm for which we could also compute a transiency index. The relation between RGC transiency and modulation index for each condition was modeled using a linear regression least-squares fit through the ON and OFF RGC population. To determine if the slope of the resulting line was statistically significant nonzero, we conducted a t test of this slope.

**Data analysis for cone photoreceptors**. We analyzed data from 11 scan fields recorded from four retinae (two mice). Each scan field was $128 \times 128$ pixels, which on the retinal surface was $94 \times 94\ \mu m^2$, $110 \times 110\ \mu m^2$, or $132 \times 132\ \mu m^2$, depending on the zoom factor used. In each scan field, we identified regions of interest (ROIs) as a group of neighboring pixels with correlated fluorescence signals in time. Only ROIs with diameters corresponding to the cone axon terminal diameter (3–7 µm) were considered for further analysis. The output signal of the ROIs (baseline normalized iGluSnFR indicator fluorescence signal), represented the changes in glutamate release at the cone terminals. A total of 931 ROIs were extracted from the 11 scan fields. Identifying the ROIs and extracting their output signal were automated using custom IGOR Pro scripts.

Within a scan field, each ROI was sampled every 256 ms (3.9 Hz sampling rate). This interval was greater than the duration of probe flashes (100 ms) in these experiments. Therefore, the measured signal of many ROIs might not capture the peak response to the probe flashes. A conventional upsampling method, such as interpolation, could also underestimate the peak response in this case. However, since all ROIs (within and across scan fields) experienced the same visual stimulus (Supplementary Fig. 1c), but were sampled at different points in time,

we temporally "stitched" the output from these ROIs. The resulting signal had a sampling interval of 2 ms, where the signal in a specific time bin was computed from ROIs sampled within that specific time window. In this "stitching" approach, we first baseline-adjusted the output signal of each ROI for a trial by subtracting the baseline activity (calculated as the average ROI output across 1 s epoch prior to the first luminance step in the trial). Then, for each 2-ms time bin, we averaged the response across those ROIs that were sampled within that time window. This resulted in an output vector of the same duration as the trial but with a sampling interval of 2 ms. The output vector was empty for time bins where no ROIs were sampled. We, therefore, convolved this output vector with a moving average filter of size 80 ms to fill in the empty time bins and to also smooth out the stitching boundaries (boundaries between time bins filled with output from different ROIs). This method gave a much better temporal resolution than conventional upsampling techniques, robustly capturing the peak for the 100 ms duration probe flash, as shown in Fig. 5b.

The "stitched" signal was then cut into snippets that captured the relevant responses to our stimulus (e.g., step followed by probe flash). For this, we used stimulation trigger signals that marked the presentation of each luminance step. Each snippet was then baseline-adjusted by subtracting the average response over 800 ms prior to the luminance step. This way the output signal was 0 prior to a luminance step (Fig. 5). The output was averaged across the three repetitions of the trial. The normalized and averaged snippets represented the cone response to a particular stimulus sequence (e.g., Fig. 5a) and were used to fit parameters for model cones as described in the next section.

**Data analysis for human psychophysics**. We analyzed eye movements in all trials and detected saccades using established methods[22]. We excluded from analysis trials in which a saccade or microsaccade happened anywhere in the interval from 200 ms before to 50 ms after a probe flash. At each flash time, we calculated the proportion of correct trials to obtain time courses of this perceptual measure. We obtained time course curves for each subject individually and then averaged it across trials and different contrasts of the luminance steps. Reduced proportion of correct trials at any flash time indicated perceptual saccadic suppression. A subset of data from four of the five subjects was used in our previous study[4]. Here we perform further analyses of the complete dataset.

We applied a two-tailed Wilcoxon rank-sum test to determine if the suppression after luminance steps differed across bright and dark probe flashes.

**Computational model of retinal ganglion cells**. To describe cone responses (Fig. 5) and RGC responses (Fig. 6) to our luminance-step paradigm, we used a phenomenological model of the retina, previously published in ref. [45].

The original model related light intensity to retinal ganglion spiking activity by three layers of processing: first, the light stimulus was passed to model cone photoreceptors. Their activity was modulated by negative feedback from model horizontal cells. Second, the output of the model cones was passed to six inner retina pathways describing retinal processing by three different ON and three different OFF bipolar cells (fast, intermediate and slow pathways). Third, the output from the model pathways were then fed into model RGCs to yield RGC spiking activity. This cascade modeled RGC spiking in response to the light stimulus passed to model cones.

The cone responses were described as

$$r(t) = \frac{\alpha_c y(t)}{[1 + \beta_c z(t)]} - h(t) \qquad (3)$$

where

$$r(t) = V(t) - V_{dark} \qquad (4)$$

$V(t)$ and $V_{dark}$ were the instantaneous and dark membrane potentials of the cone, respectively, $h(t)$ was the feedback signal from the horizontal cell, and $\alpha_c$ and $\beta_c$ were numerical factors. The time-varying functions $y(t)$ and $z(t)$, were related to light input through linear convolution, as

$$y(t) = \int_{-\infty}^{t} K_y(t - t')I(t')dt' \qquad (5)$$

$$z(t) = \int_{-\infty}^{t} K_z(t - t')I(t')dt' \qquad (6)$$

where $I(t)$ was the incident light intensity (or, more precisely, $R^* s^{-1}$). The kernels describing the cone response were given by

$$K_y(t) = \frac{t}{\tau_y} \frac{e^{-\frac{t}{\tau_y}}}{\tau_y} \qquad (7)$$

and

$$K_z(t) = \gamma K_y(t) + (1 - \gamma) \frac{t}{\tau_z} \frac{e^{-\frac{t}{\tau_z}}}{\tau_z} \qquad (8)$$

where $\tau_z$ was larger than $\tau_y$, and $0 \le \gamma \le 1$ ensured proper normalization. Note that $\int_0^\infty dt' K_y(t - t') = 1$ for all filters. The response of the horizontal cell was described

by

$$h(t) = \alpha_c \int_{-\infty}^{t} K_h(t - t') r(t') dt' \quad (9)$$

with

$$K_h(t) = \frac{t}{\tau_h} \frac{e^{-\frac{t}{\tau_h}}}{\tau_h} \quad (10)$$

Here, instead of the cone model parameters used in the published model[45], we re-fitted the parameters of the model cone to reflect our measured data of cone output (Fig. 5b) which yielded faithful fits (Supplementary Fig. 11) to our experimentally measured cone responses.

All the parameters of the outer retina component (Eqs. (3) and (10)) of our circuit model were fit once to cone responses (Fig. 5b and Supplementary Fig. 11) and then kept unchanged for all simulations reported in Fig. 6. The fitted values are given as follows (original values[45] are reported in brackets):

$$\alpha_c = -3.342 * 10^{-5} \left(-9.602 * 10^{-6}\right),$$
$$\beta_c = -1.273 * 10^{-6} \left(-1.148 * 10^{-5}\right),$$
$$\gamma = 0.842 (0.764), \alpha_h = 0.016 (0.177),$$
$$\tau_y = 48.98\ ms (50.64), \tau_z = 200\ ms (576.9),$$
$$\tau_h = 1.232 * 10^3 ms (371).$$

The small $\alpha_h$ suggested that in our cone recordings, horizontal cell feedback had only a minor effect.

In ref. [45], three different retinal pathways were modeled according to

$$b_{p,k}(t) = \left\lfloor -1^k \left( \int_{-\infty}^{t} K_p(t - t') V(t') dt' - \theta_{p,k} \right) \right\rfloor \quad (11)$$

where $p = 1, 2, 3$ labeled the pathway based on its response properties (1 = fast, 2 = intermediate, 3 = slow), $k = 0$ for OFF pathways and $k = 1$ for ON pathways.

$$\lfloor x \rfloor = \begin{cases} 0, & x<0 \\ x, & x \geq 0 \end{cases} \quad (12)$$

was a thresholding nonlinearity, and $\theta_{p,k}$ acted as a threshold.

The main difference between the pathways was the temporal characteristics of the filters $K_p$. In this study, we wanted to smoothly vary the transiency of the model ganglion cells. To this end, we based our bipolar pathway on the fast bipolar pathway ($p = 1$) and modified its temporal characteristics to make it less transient. $K_1$ represented a high-pass filter which took the derivative of the cone potential on the order of 1 ms. We obtained $K_1$ by convolving the high-pass filter of the form

$$G(t) = \sin\left(\frac{\pi t}{\mu}\right) \frac{1}{\sqrt{2\pi\sigma}} e^{-\frac{1}{2}\left(\frac{t-\mu}{\sigma}\right)^2}, with\ \mu = 3\ ms, \sigma = 1\ ms \quad (13)$$

with an exponential function.

$$K_1(t) = \int_{-\infty}^{t} \left(e^{-\frac{t-t'}{\tau_d}}\right) G(t') dt' \quad (14)$$

Higher values of the time constant, $\tau_d$, of the exponential function decreased the pathway transiency. We set $\tau_d = 0.5\ ms$ as the default transient pathway (Fig. 6a–d). The filter transiency values shown in Fig. 6g were obtained by normalizing $15\ ms \geq \tau_d \geq 0.5\ ms$ in the range 0 to 1, with 0 being less transient. In the original model[45], $K_1(t) = G(t)$.

The threshold $\theta_{1,k}$ was set to $-1^k \cdot 0.1$, except in Fig. 6h, where we varied its value between 0 and 1, while keeping $\tau_d$ fixed at a value of 15 ms (or 0 in the normalized scale). This change in nonlinearity was another way to change response properties of the pathway. Note that the threshold values are not on a normalized scale.

$b_{1,k}(t)$, the output of fast inner retina models was used as the input to the model RGCs used in this study. The spiking rate of the model RGC was obtained as the thresholded input and a temporally coarse version of the input's derivative,

$$R_{1,k}(t) = \left\lfloor (1 - \alpha) b_{1,k}(t) + \alpha \left( \int_{-\infty}^{t} K(t - t') b_{1,k}(t') dt' \right) - \theta \right\rfloor \quad (15)$$

where $K(t)$ was a biphasic filter similar in its form to $G(t)$. The threshold, $\theta$, was a multiple of the peak response to any given input. We used the same parameters for the inner retina component as the ones described in the published model[45]:

(i) Transient OFF, $R_{1,0}(t)$ : $\alpha = 0, \theta = 0$;
(ii) Transient ON, $R_{1,1}(t)$ : $\alpha = 0, \theta = 0$;

All simulations were computed with a 1 ms sampling interval.

Using the above cascade, we calculated model RGC's spike rate in response to the recorded cone output when subjected to "luminance steps alone" and "luminance steps followed by probe flash" stimuli (Fig. 6a, b). Similar to real RGC analysis, we calculated the "flash-induced responses" (Fig. 6c) by subtracting "response to luminance steps alone" (Fig. 6a) from "response to luminance steps followed by probe flash" (Fig. 6b). We then calculated the modulation index also in the same way as for the real RGCs: $(r_d - r_b)/(r_d + r_b)$ where $r_d$ was the peak spiking rate of the flash-induced response for the flash presented with delay $d$ from the

luminance step, and $r_b$ was the peak firing rate for the baseline flash-induced response (flash at 2000 ms).

As a control, we replaced the model cone responses with the experimentally acquired cone responses (Fig. 5) thus forming a hybrid model. Before passing the cone response to the downstream model pathways, we passed it through a low-pass filter to further smooth the fluctuations at the stitching boundaries in order to avoid discontinuities in the calculation of its temporal derivative. For this smoothing, we convolved the cone output with a moving average filter of size 40 ms. Here, the model ON RGC responses (Supplementary Fig. 12a, b, columns 1–2) were calculated using the cone responses to bright probe flashes (Fig. 5a, b, columns 1–2), and OFF RGC responses (Supplementary Fig. 12a, b, columns 3–4) were calculated using cone responses to dark probe flashes (Fig. 5a, b, columns 3–4). The resulting hybrid model RGC responses (Supplementary Fig. 12) were consistent with the pure model responses shown in Fig. 6.

**Statistics and reproducibility**. We applied statistical tests both at the level of individual RGCs and at the level of the population to determine whether the response to a probe flash following a saccade or a luminance step was significantly modulated. At the individual cell level, we determined whether the modulation index for a probe flash presented at a certain delay was significantly different from 0 (i.e., "Is the response of this cell modulated by the 'saccade'?"). For this, we performed a one-tailed sign test of the null hypothesis that the 39 individual modulation indices (or its subset in case weak sequences were discarded, as described above) came from a distribution with zero median. The alternative hypothesis was that the median was below (for negative modulation index) or above (for positive modulation index) zero. The modulation index was considered significant (i.e., the flash response was modulated by the saccade) at $P<0.05$. However, we did not consider cells significantly modulated if the test had a power $(1-\beta)$ smaller than 0.8, which could happen if we previously had to exclude too many sequences ($N \leq 39$). At the population level, we determined whether the retinal output as a whole was modulated by saccades. For this, we performed a two-tailed Wilcoxon signed-rank test of the null hypothesis that the median of the distribution of modulation indices did not differ from 0. Lastly, we tested whether the population modulation index was significantly different across populations of ON and OFF RGCs or across different paradigms. For this, we performed a two-tailed Wilcoxon rank-sum test of the null hypothesis that the median of the distribution of modulation indices did not differ across the two populations being tested.

Since the modulation index was based on responses to the brief probe flashes, it could only be computed for RGCs that did respond to these brief flash stimuli. In our analysis, we included all such RGCs. Of the spike sorted RGCs across all paradigms, we included: 2002 of 3510 in mice (47 retinae); 228 of 376 in pigs (pieces from 12 retinae); and 15 of 57 in macaque (1 retina).

In human psychophysics experiments, we applied a two-tailed Wilcoxon rank-sum test to determine if the suppression after luminance steps differed across bright and dark probe flashes ($N = 5$ subjects).

All data analyses were performed in MATLAB (The MathWorks Inc).

**Reporting summary**. Further information on research design is available in the Nature Research Reporting Summary linked to this article.

## Data availability

The source data underlying the figures in this manuscript are available in the public repository https://github.com/saadidrees/saccadic-suppression with identifier https://doi.org/10.5281/ZENODO.6562460 (ref. [79]). All other data are available upon reasonable request.

## Code availability

The code and source data underlying the figures in this manuscript are available in the public repository https://github.com/saadidrees/saccadic-suppression with identifier https://doi.org/10.5281/ZENODO.6562460 (ref. [79]). All other code used for analysis and computational modeling is available upon reasonable request.

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

## Acknowledgements

We thank Elisabeth Gustafsson for providing technical support; Andreas Hierlemann for providing the HiDens CMOS MEA system and helping establish our high-density MEA recordings; Roland Diggelmann for helping in setting up the pipeline (including providing code) for automatic spike sorting of high-density MEA recordings; Martin Schenk at the Department of Experimental Surgery, Tübingen, for providing us domestic pig eyes. Our work was supported by funds of the Deutsche Forschungsgemeinschaft (DFG) to the Werner Reichardt Center for Integrative Neuroscience (EXC 307) and to TAM (MU3792/1-1 and MU3792/3-1). T.A.M. received support from the Tistou and Charlotte Kerstan Foundation. S.I., T.A.M., and Z.M.H. were also supported by an intra-mural funding program (Project 2013-05) of the Werner Reichardt Center for Integrative Neuroscience. F.F. was supported by a Swiss National Science Foundation Ambizione grant (PZ00P3_167989) and Swiss National Science Foundation Eccellenza grant (PCEFP3_187001). M.P.B. and Z.M.H. were further funded by the SFB 1233 on Robust Vision (DFG, project number 276693517). K.F. was supported by the Bundesministerium für Bildung und Forschung (BMBF, 01GQ1002) and the Max Planck Gesellschaft (MPG, M.FE.A.KYBE0004).

## Author contributions

T.A.M. conceptualized the study; S.I. and T.A.M. designed the overall study; F.F. developed and supervised computational modeling; S.I., M.P.B., A.K., K.F., Z.M.H., F.F., and T.A.M. designed experiments; S.I. performed ex vivo mouse and pig retina experiments; A.K. performed ex vivo macaque retina experiment; M.P.B. performed human psychophysics experiments; K.F. performed viral injections; M.K. performed cone imaging experiments; S.I. and F.F. implemented computational model; S.I., M.P.B., M.K., T.S., A.K., K.F., Z.M.H., F.F., and T.A.M. analyzed and interpreted the data and wrote the manuscript.

## Funding

## Competing interests

The authors declare no competing interests.
