## [Peer Review File · Communications Biology]

REVIEWERS' COMMENTS:

Reviewer #1 (Remarks to the Author):

The manuscript by Idrees et al. has much improved and clarified. I only have some minor comments left.

The model of ganglion cell responses used to analyze the effects of model transiency and of the threshold in the nonlinearity is really difficult to parse. I would be helpful, for example, if examples of the filters could be shown. (E.g. for transiency parameter = 0, 0.5, 1 or something like this.)

Also, why is the nonlinearity transiency parameter called this way? Does it relate to transiency? And maybe explain in the main text (around lines 455) in what sense it is made "more strict". I guess more strict corresponds to higher threshold?

In the Methods section, I was confused by the small time scales ($\mu=3\text{ms}$, $\sigma=1\text{ms}$), which should lead to very brief and rapidly oscillating filters, whereas Fig. 6e indicates that the applied stimuli are in the range of 100 ms. And τ_d in line 1839 comes without units. Finally, Equation 12 contains a sine function within a sine function. Is this a typo?

Line 428: I didn't understand in what sense the rate of change in cone output becomes "much smaller". Maybe explain what you are comparing here.

In the new section on the apparent pre-saccadic suppression, I was wondering whether one should think of the response to non-preferred contrast as a response to the onset of the flash (thus relating to ON-OFF cells) or whether this is rather a rebound response to the offset of the flash. Maybe the authors want to at least mention these possibilities, as they may be important when considering potential mechanisms.

Responses to Reviewer Comments on:

“Suppression without inhibition: How retinal computation contributes to saccadic suppression”
Idrees, Baumann, Korympidou, Schubert, Kling, K Franke, Hafed, F Franke*, Münch*

We thank the reviewer for their insightful comments to help finalize our manuscript.

In what follows, we provide specific responses to the reviewer comments (colored in blue text), which we have incorporated into the final version of this manuscript.

The manuscript by Idrees et al. has much improved and clarified. I only have some minor comments left.

We thank the reviewer for this encouraging remark.

The model of ganglion cell responses used to analyze the effects of model transiency and of the threshold in the nonlinearity is really difficult to parse. I would be helpful, for example, if examples of the filters could be shown. (E.g. for transiency parameter = 0, 0.5, 1 or something like this.)

We have added a new figure (Supplementary Fig. 13) showing the different filters and effect of different nonlinearities on the cone output.

Also, why is the nonlinearity transiency parameter called this way? Does it relate to transiency? And maybe explain in the main text (around lines 455) in what sense it is made “more strict”. I guess more strict corresponds to higher threshold?

As pointed out by the reviewer, more strict indeed corresponds to higher threshold values. We have added this explanation in the main text (lines 451-454). We have also simplified the terminology by calling it a thresholding nonlinearity. Similarly, the filter transiency parameter is now referred to as the filter transiency.

In the Methods section, I was confused by the small time scales ($\mu=3\text{ms}$, $\sigma=1\text{ms}$), which should lead to very brief and rapidly oscillating filters, whereas Fig. 6e indicates that the applied stimuli are in the range of 100 ms. And τ_d in line 1839 comes without units. Finally, Equation 12 contains a sine function within a sine function. Is this a typo?

We thank the reviewer for pointing out the missing units and the typo. τ_d is in ms and we have added this unit in the Methods section (lines 1635-1637). We have also corrected the typo in Equation 12 (now Equation 13). Regarding the time scales, we have added a new figure (Supplementary Fig. 13) showing the shape of different filters.

Line 428: I didn't understand in what sense the rate of change in cone output becomes “much smaller”. Maybe explain what you are comparing here.

We are comparing the rate of change in cone output of a flash presented immediately after a luminance step with a flash presented long after the luminance step. We have edited the text to make this comparison explicit (lines 423-426)

In the new section on the apparent pre-saccadic suppression, I was wondering whether one should think of the response to non-preferred contrast as a response to the onset of the flash (thus relating to ON-OFF cells) or whether this is rather a rebound response to the offset of the flash. Maybe the authors want to at least mention these possibilities, as they may be important when considering potential mechanisms.

In comparison to preferred contrast flashes, the responses to non-preferred contrast have a much higher latency with respect to both the onset and offset of the flash. The origins, mechanisms and the role of these delayed responses are not well studied, which we explicitly state in lines 658-662.